# Antibiotic thermorubin tethers ribosomal subunits and impedes A-site interactions to perturb protein synthesis in bacteria

Narayan Prasad Parajuli ◉[1], Andrew Emmerich ◉[1], Chandra Sekhar Mandava[1], Michael Y. Pavlov[1] & Suparna Sanyal ◉[1] ✉

Thermorubin (THB) is a long-known broad-spectrum ribosome-targeting antibiotic, but the molecular mechanism of its action was unclear. Here, our precise fast-kinetics assays in a reconstituted *Escherichia coli* translation system and 1.96 Å resolution cryo-EM structure of THB-bound 70S ribosome with mRNA and initiator tRNA, independently suggest that THB binding at the intersubunit bridge B2a near decoding center of the ribosome interferes with the binding of A-site substrates aminoacyl-tRNAs and class-I release factors, thereby inhibiting elongation and termination steps of bacterial translation. Furthermore, THB acts as an anti-dissociation agent that tethers the ribosomal subunits and blocks ribosome recycling, subsequently reducing the pool of active ribosomes. Our results show that THB does not inhibit translation initiation as proposed earlier and provide a complete mechanism of how THB perturbs bacterial protein synthesis. This in-depth characterization will hopefully spur efforts toward the design of THB analogs with improved solubility and effectivity against multidrug-resistant bacteria.

The global prevalence of antimicrobial resistance (AMR) among bacterial pathogens coupled with a barren antibiotic discovery pipeline[1] represents an imminent public health crisis[2]. There is an urgent need for the development of novel antibiotics with unique target sites to circumvent the ever-evolving problem of AMR among bacterial pathogens. However, the discovery of new antibiotics is rather challenging[1]. In that context, previously discovered antimicrobial compounds that have been neglected so far due to certain limitations can become attractive alternatives[3]. Understanding the molecular basis of action of these compounds could provide important clues for rational modification of the existing compounds[4] or lead to the total synthesis of newer compounds to combat antibiotic resistance[5].

Thermorubin (THB) is a naturally occurring antibiotic produced by *Thermoactinomyces antibioticus*, a thermophilic actinomycete[6,7]. Originally discovered in 1964 as a broad-spectrum antibiotic, THB is effective against Gram-positive and Gram-negative bacteria with minimum inhibitory concentration (MIC) ranging from 0.025–0.05 μg ml⁻¹[8]. It is inactive against protozoans, yeast,

filamentous fungi, and other higher eukaryotes[6]. Chemically, THB is a unique example of an aromatic anthracenopyranone metabolite, with anthraquinone moiety connected to an orthohydroxyphenyl moiety (Fig. 1a) presumably produced by the condensation of its polyketide precursors[9]. The linear tetracycle in THB has some resemblance to that of tetracyclines, but in contrast, it is entirely aromatic without a single chiral center[7]. The highly conjugated structure with multiple aromatic rings makes THB strongly hydrophobic and poorly soluble in aqueous media[8]. This is why regardless of its potent antibacterial effect with low toxicity (Therapeutic Index > 200), THB has seldom been used for the treatment of infectious diseases[8,10]. Thus, it has been neglected for a long time and the precise mechanism of THB action remained elusive. This is probably also the reason for limited efforts towards design of the THB analogs with improved solubility.

THB is a bacteriostatic antibiotic; early investigations using *Escherichia coli* uncovered its selective inhibitory effects on protein synthesis[11]. Subsequently, it was discovered that THB binds to 70S ribosome very tightly ($K_d \approx 20$ nM) with an affinity ~100-folds higher

[1]Department of Cell and Molecular Biology, Uppsala University, Uppsala SE-75124, Sweden. ✉e-mail: suparna.sanyal@icm.uu.se

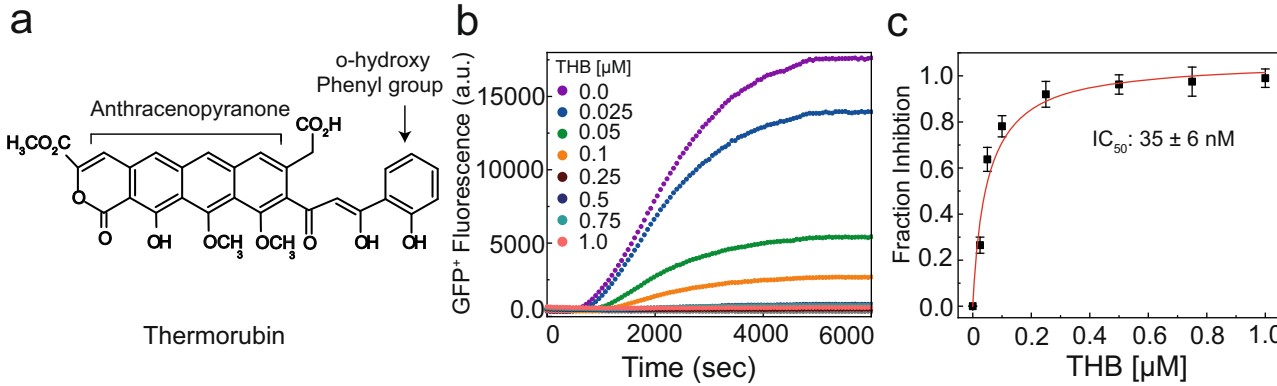

**Fig. 1 | THB and its effect on protein synthesis in bacteria. a** Chemical structure of antibiotic Thermorubin A (created using ChemSketch). **b** Inhibitory effects of increasing concentrations of THB on in vitro GFP+ synthesis in an *E. coli* based reconstituted translation system. **c** The fraction of THB-induced inhibition of GFP+ synthesis estimated from (**b**). The solid lines in (**c**) represent the hyperbolic fit of the data from which half-maximal inhibitory concentration (IC50) was obtained. Experiments were conducted in duplicates. Error bars indicate SEM of the data.

than to individual ribosomal subunits, suggesting a single high-affinity site for THB binding formed by the subunit interface[12]. This suggestion was lately confirmed by a crystal structure of THB-bound 70S ribosome from the bacterium *Thermus thermophilus*[13] which located its binding to the ribosomal intersubunit bridge B2a formed by the conserved nucleotides of 16S and 23S rRNAs of the small and large ribosomal subunits, respectively (PDB 4V8A)[13]. This binding site partially overlaps with that of the other clinically important antibiotics such as aminoglycosides and tuberactinomycins[14,15]. The linear tetracyclic moiety of THB stacks between C1409-G1491 base pair within helix 44 (h44) of 16S rRNA and A1913 within helix 69 (H69) of 23S rRNA, while its orthohydroxyphenyl moiety stacks upon U1915 of H69[13]. The nucleotide corresponding to C1409 in the eukaryotic ribosome is not base paired, which might prevent stacking of the tetracyclic moiety of THB, explaining its selectivity towards bacterial ribosomes[3,8,11]. Moreover, THB binding to the bridge B2a induces C1914 to flip out from H69 and adopt a conformation that can be incompatible with binding of the A-site ligands[13]. However, early biochemical studies claimed that THB inhibits the binding of initiator tRNA to the P-site in an initiation factor-dependent manner[11,12]. Noteworthy, these studies on THB-induced translation inhibition relied largely on S30 cell-extract[11] or poly-uracil (poly-U) mRNA-based translation assays[10,12]. Although bridge B2a harbors nucleotides interacting with IF1 and IF2[16], it remained unclear how the THB-induced conformational dynamics could interfere with the activity of the initiation factors in bringing fMet-tRNA[fMet] to the P-site of the 30S subunit and stabilizing it there during translation initiation. Moreover, in the pivotal biochemical experiments[10,12] that led to the conclusion that THB inhibits the initiation of protein synthesis, more than 40% of the ribosomes still contained fMet-tRNA[fMet] even at a saturating THB concentration[12]. The uncertainties about the step of translation THB actually targets prompted us to undertake a detailed investigation of structure-function mechanisms of THB inhibition of bacterial protein synthesis.

In this work, we use a reconstituted in vitro translation system assembled from purified translation components of *E. coli*[17] to clarify the effects of THB on initiation, elongation, termination, and ribosome recycling steps of translation. The results of our fast kinetics experiments combined with our cryo-EM structure of the THB-containing ribosome initiation complex at 1.96 Å resolution demonstrate that the ribosome-bound THB has virtually no effect on initiation. Instead, THB impedes A-site delivery and accommodation of aa-tRNA, thereby impairing elongation. It also inhibits severely A-site binding of the class I release factors (RF) thereby inhibiting translation termination. Furthermore, THB acts as a 'glue' that tethers the ribosomal subunits together and inhibits splitting of the post-termination ribosome into

subunits. Collectively, these findings refute the earlier notion of THB as an initiation inhibitor and provide comprehensive quantitative insights into the mechanism of translation inhibition by THB.

## Results

### THB strongly inhibits in vitro protein synthesis in bacteria

We first studied the effect of THB on synthesis of a full-length protein with multiple turnover of ribosomes and translation factors. For that, we used an *E. coli* based fully reconstituted translation system[18] and followed the synthesis of fluorescent reporter protein GFP+ [19]. THB reduced the production of GFP+ in a concentration-dependent manner (Fig. 1b). The half-maximal inhibitory concentration (IC50) estimated from the mid-point of the hyperbolic fit was 35 ± 6 nM (Fig. 1c) and the synthesis of GFP+ essentially ceased at 250 nM of THB. Having confirmed the strong inhibitory effect of THB on bacterial translation in our in vitro system, we further studied THB's effects on the initiation, elongation, termination, and recycling stages of translation.

### THB does not interfere with initiator tRNA binding and subunit association during translation initiation

The inhibition of the initiation factor-dependent delivery of initiator tRNA into the P-site of the ribosome has been claimed earlier as the focal point of THB inhibition of bacterial protein synthesis[12]. We re-investigated this claim using the native substrates of initiation: fMet-tRNA and native-like XR7 mRNA that had a strong Shine-Dalgarno sequence followed by a short coding sequence for Met-Phe-Phe-stop codons (see Methods for the mRNA sequence). Here, we studied the effect of THB on subunit association, occupancy of fMet-tRNA[fMet] in the 70S initiation complex (70S IC), and puromycin reactivity of fMet-tRNA[fMet] in 70S IC.

In the subunit association assay (illustrated in Supplementary Fig. 1), 30S pre-initiation complex (30S pre-IC) containing 30S subunit bound to mRNA, fMet-tRNA[fMet], IF1, IF2, and IF3 (see "Methods" and Supplementary Note 1 for the details) was rapidly mixed with 50S subunit in a stopped-flow instrument and the formation of 70S IC was monitored by Rayleigh light scattering (Fig. 2a)[20]. The second-order association rate constant ($k_a$) for subunit association was estimated using Supplementary Eq. (9) as in ref. [21]. Interestingly, the $k_a$ for 70S IC formation without or with THB was similar (11 ± 2.1 $\mu M^{-1} s^{-1}$ vs 9.6 ± 1.5 $\mu M^{-1} s^{-1}$) demonstrating that THB had no inhibitory effect on subunit association. THB also showed negligible to no effect on association of 50S with naked 30S (Supplementary Fig. 2a) and 30S pre-IC containing no IF3 (Supplementary Fig. 2b).

Next, we used the nitrocellulose filter-binding (NC) assay to estimate the extent of f[³H]fMet-tRNA[fMet] binding to the

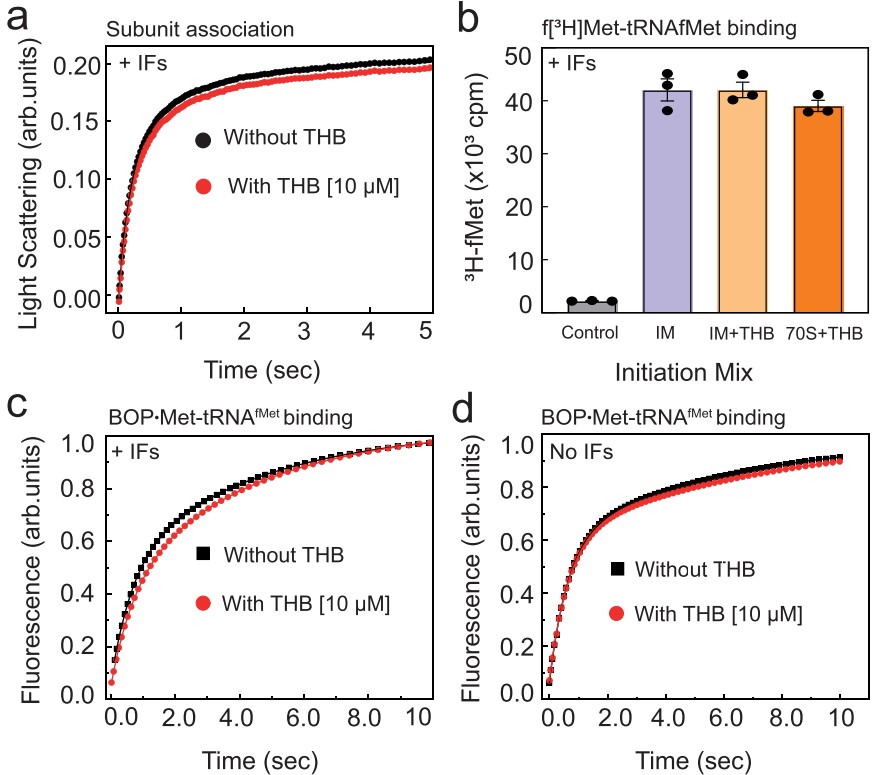

**Fig. 2 | Effects of THB on subunit association and initiator tRNA binding during translation initiation. a** Time course of 50S association to 30S pre-IC in the presence of all three IFs, with or without THB. Subunit association was monitored as an increase in Rayleigh light scattering at 365 nm. Solid lines are the fits of data to complex formation equation (Supplementary Eq. 9). **b** The bars represent the extent of bound f[³H]Met-tRNA$^{fMet}$ to the mRNA-programmed 70S ribosomes in the presence of all IFs after incubation at 37 °C for 15 min ($n = 3$). THB was either added after 70S IC formation (marked IM + THB) or was pre-incubated with 70S for 5 min before addition of mRNA, f[³H]Met-tRNA$^{fMet}$ and the IFs (marked 70S + THB). **c** Real-time fluorescence traces of BOP•Met-tRNA$^{fMet}$ binding to mRNA-programmed 70S ribosomes in the presence and (**d**) absence of IFs. THB was pre-incubated with the ribosomes before mixing with BOP•Met-tRNA$^{fMet}$ in a stopped-flow instrument. Data were fitted to complex formation equation (Supplementary Eq. 8) to estimate the effective rate of binding without ($k_b$) and with THB ($k_{b^*}$). Experiments were conducted in triplicates and error bars indicate SEM.

mRNA-programmed 70S ribosome in the presence or absence of THB. The 70S IC was prepared by incubating 70S ribosome, XR7 mRNA, f[³H]Met-tRNA$^{fMet}$, IF1, IF2 and IF3. The extent of f[³H]Met-tRNA$^{fMet}$ binding to the 70S IC was measured from retention of f[³H]Met-tRNA on the nitrocellulose membrane (see Methods for details). Comparison of the amounts of 70S-ribosome-bound f[³H]Met-tRNA$^{fMet}$, in the presence and absence of THB, revealed no difference in the 70S IC formation (Fig. 2b). Even a long pre-incubation of THB with 70S ribosome before addition of fMet-tRNA$^{fMet}$ and mRNA showed no effect on 70S IC formation (Fig. 2b). This result is clearly contrary to the earlier result of a similar experiment in which a model poly (A, U, G) mRNA mimic was used[12] instead of the native-like mRNA used here. We further checked puromycin reactivity of f[³H]Met-tRNA$^{fMet}$ in the 70S IC formed in the presence of THB to confirm its P-site location and activity. THB showed no effect on puromycin reaction either, thus implying that THB not only allows fMet-tRNA$^{fMet}$ binding to the P-site, but also does not interfere with the formation of elongation-competent 70S IC and the subsequent peptidyl transfer reaction (Supplementary Fig. 3).

We further studied the real-time kinetics of BOP•Met-tRNA$^{fMet}$ binding to the mRNA-programmed 70S ribosome (as illustrated in Supplementary Fig. 1) by monitoring the increase in BOP fluorescence in a stopped-flow instrument (Fig. 2c). The effective rate ($k_b$) of BOP•Met-tRNA$^{fMet}$ binding was estimated by fitting the BOP•Met fluorescence traces to Supplementary Eq. (8). In the presence of all three IFs, BOP•Met-tRNA$^{fMet}$ binding to mRNA-programmed 70S proceeded with $k_b = 0.27 \pm 0.02\ s^{-1}$; that did not change appreciably upon addition of THB *i.e.*, $k_{b^*} = 0.24 \pm 0.02\ s^{-1}$ (Fig. 2c). Similarly, THB did not

affect BOP•Met-tRNA$^{fMet}$ binding to 70S-mRNA without IFs (Fig. 2d). Taken together, these results show that THB has no effect on the binding of the initiator tRNA to the 70S ribosome and it does not inhibit translation initiation.

## THB interferes with the ternary complex binding and subsequent aa-tRNA accommodation during peptide elongation

Our observation that THB does not interfere with the initiator tRNA binding into the P-site of the ribosome suggests that THB should act on the subsequent steps of mRNA translation. Since THB binds in the vicinity of the A-site of the ribosome, we reasoned that the drug might impede the elongation cycle of the ribosome that consists of aminoacyl (aa)-tRNA delivery to the A-site of the ribosome by EF-Tu•GTP, subsequent aa-tRNA accommodation, peptidyl transfer reaction in the peptidyl transferase center (PTC) and EF-G mediated peptidyl-tRNA translocation from the A-site to the P-site of the ribosome (see Supplementary Fig. 4a for the kinetic scheme of ribosome elongation cycle).

To pinpoint the exact step(s) of the elongation cycle affected by THB, we reacted ternary complexes ($T_3$) containing EF-Tu, [³H]GTP, and Phe-tRNA$^{Phe}$ with an excess of mRNA-programmed 70S ribosome carrying f[³H]Met-tRNA$^{fMet}$ in the P-site and a cognate (UUC) codon in the A-site in a quench-flow instrument. First, we monitored the time course of GTP hydrolysis on EF-Tu, which takes place concomitantly with aa-tRNA delivery to the A-site of the ribosome upon codon recognition (Fig. 3a). In parallel, we followed dipeptide (fMet-Phe) formation, which includes, in addition to the delivery, aa-tRNA accommodation and subsequent peptidyl transfer reaction (Fig. 3b).

To ensure single cycle conditions in these experiments we omitted EF-Ts, a GDP to GTP exchange factor for EF-Tu, so that the $T_3$ hydrolysis-product EF-Tu•GDP cannot be converted back to $T_3$. By fitting GTP hydrolysis data (Fig. 3a) with single-exponential function (Supplementary Eq. 38), and dipeptide formation data (Fig. 3b) with double-exponential function (Supplementary Eq. 39) we determined the mean times of GTP hydrolysis on EF-Tu ($\tau_{GTP}$) and dipeptide formation ($\tau_{Dip}$), respectively. Further, the sum of mean times of aa-tRNA accommodation and subsequent peptidyl transfer reaction (together referred as $\tau_{A/PT}$) was estimated by subtracting $\tau_{GTP}$ from $\tau_{Dip}$[22] (see Supplementary Note 2 for details).

In the absence of THB $\tau_{GTP}$ was $9.5 \pm 2.5$ ms; that increased to $192 \pm 12$ ms in the presence of THB (Fig. 3a) demonstrating that THB significantly delays delivery of aa-tRNA by $T_3$. Further, $\tau_{Dip} \approx 19 \pm 4$ ms in the absence of THB increased to $472 \pm 18$ ms in its presence (Fig. 3b). Thus, the $\tau_{A/PT} \approx 9.5 \pm 3.2$ ms estimated as $\tau_{Dip} - \tau_{GTP}$ in the absence of THB increased to more than $280 \pm 27$ ms in its presence. Since our puromycin assay (Supplementary Fig. 3) did not show any defect caused by THB on peptidyl transfer reaction, we reasoned that the additional delay of $280 \pm 27$ ms after aa-tRNA delivery is due to impairment of aa-tRNA accommodation into the PTC in the presence of THB. Interestingly, when the concentration of ribosome was doubled, $\tau_{GTP}$ in the presence of THB reduced two-fold, from $192 \pm 12$ ms to $104 \pm 7$ ms, indicating that THB affects the efficiency ($k_{cat}/K_M$) of the EF-Tu dependent aa-tRNA delivery reaction (Supplementary Fig. 5). From $\tau_{GTP}$ data we estimated a 20-fold reduction in $k_{cat}/K_M$ of the aa-tRNA delivery reaction from $100\ \mu M^{-1} s^{-1}$ in THB's absence to $5\ \mu M^{-1} s^{-1}$ in its presence (Fig. 3a).

Our results show that the amount of dipeptide formed with THB was approximately half compared to that without it (Fig. 3b) despite the same extent of GTP hydrolyzed in the absence and presence of THB (Fig. 3a). Thus, in addition to inhibiting aa-tRNA delivery THB also induces aa-tRNA drop-off from the A site. However, when $T_3$ was added in large excess over ribosome, dipeptide formation proceeded to completion (Fig. 3c). It means that the aa-tRNA drop-off effect of THB can be overcome by repeated binding of $T_3$ to aa-tRNA-free ribosomes (see Supplementary Note 2 for details).

Importantly, when THB was added together with $T_3$ to the 70S IC no discernible inhibition of dipeptide formation rate was observed (Supplementary Fig. 6). It suggests that THB binding to the ribosome occurs on a much slower time scale than that of $T_3$. The slow binding of THB was further confirmed by directly measuring the second-order rate constant ($k_{THB}$) for THB binding to the 70S ribosome in stopped-flow, monitored by the change in THB fluorescence at 340 nm[10]. The value of $k_{THB}$ ~ $0.03\ \mu M^{-1} s^{-1}$ (Supplementary Fig. 7) is indeed much smaller than that $k_{cat}/K_M$ of ~$100\ \mu M^{-1} s^{-1}$ for binding of $T_3$ to the ribosome in the absence of THB.

We have also used chase experiment (illustrated in Supplementary Fig. 8 and discussed in Supplementary Note 1) to obtain an upper bound ~$0.003\ s^{-1}$ for THB dissociation rate constant $q_{THB}$ from 70S ribosome and 70S IC (Supplementary Fig. 9), indicating that spontaneous dissociation of THB from the ribosome takes at least 5 min. We next studied the effect of THB on dipeptide formation in the experimental setup where 30S pre-IC containing f[$^3$H]Met-tRNA$^{fMet}$ and all IFs were rapidly mixed with 50S subunit and $T_3$ in a quench-flow instrument. Here, dipeptide formation required 50S association with 30S pre-IC to form 70S IC before $T_3$ binding and peptide bond formation. THB addition to both 30S pre-IC and 50S/$T_3$ mixes before mixing them in the quench-flow instrument showed only a negligible effect of THB on dipeptide formation (Supplementary Fig. 10). This demonstrates that 70S IC formed in this reaction do not contain THB, implying that THB is a weak binder of the individual subunits. This result also demonstrates that a subsequent fast $T_3$ binding to THB-free 70S IC outcompetes slow THB binding to it.

## THB does not affect the accuracy of aa-tRNA selection by the ribosome

To check if THB affects the accuracy of tRNA selection we conducted GTP-hydrolysis experiments using ribosome programmed with near-cognate (CUC) codon in the A-site. We observed about 20-fold reduction in the $k_{cat}/K_M$ values of near-cognate GTPase reaction upon THB addition, from $0.75\ \mu M^{-1} s^{-1}$ in the absence of THB to $0.037\ \mu M^{-1} s^{-1}$ in its presence. Interestingly, this is similar to 20-fold reduction of $k_{cat}/K_M$ for GTP hydrolysis by THB for the cognate case (Fig. 3a, Supplementary Fig. 11). Thus, we conclude that THB does not affect the accuracy of initial aa-tRNA selection on the ribosome.

## THB has no effect on EF-G-dependent translocation

We next asked if the THB-induced inhibition of elongation is limited to the first peptide bond formation or persists through the whole elongation cycle. To this end, we monitored the kinetics of tri- and tetra-peptide formation in a quench-flow instrument. As in case of dipeptide formation experiments, the 70S IC carried f[$^3$H]Met-tRNA$^{fMet}$ in the P-site but was programed with Met-Phe-Phe-stop or Met-Phe-Phe-Phe-stop mRNA coding for fMFF tri- or fMFFF tetrapeptide. These 70S ICs were mixed with a large excess of cognate ternary complexes EF-Tu•GTP•Phe-tRNA$^{Phe}$ ($T_3$) and elongation factor G (EF-G) that translocate peptidyl-tRNA from the A- to P-site of the ribosomes. After rapid mixing in a quench-flow instrument, the time evolution of di- and tri-peptide was measured in the case of Met-Phe- Phe-stop mRNA and that of di-, tri- and tetra-peptide in the case of Met-Phe-Phe-Phe-stop mRNA. From these experiments, the mean times of di- and tripeptide formation could be extracted in the first case and the mean times of di-, tri- and tetra- peptide formation in the second case (see Supplementary Note 2 and references therein for details). The mean time for tripeptide formation ($\tau_{Trip}$) $101 \pm 9$ ms, measured with 5 μM EF-G in the absence of THB, increased to $1024 \pm 35$ ms in the presence of THB (Fig. 3d, e, Supplementary Fig. 12a). Notably, $\tau_{Trip}$ with 20 μM EF-G was almost same as with 5 μM EF-G in the presence of THB (Supplementary Fig. 12b). Since tripeptide formation involves two peptidyl transfer events and one EF-G catalyzed translocation event (illustrated in Supplementary Fig. 4c), the mean time of translocation $\tau_{Trans}$ could be estimated by subtracting the mean times of two dipeptide formation (2x $\tau_{Dip}$) events from the measured $\tau_{Trip}$ (see Supplementary Eq. (52) and refs. [22,23] for details). Surprisingly, the estimated $\tau_{Trans}$ was similar both in the presence and absence of THB, i.e. $63 \pm 9$ ms, indicating that THB does not affect EF-G catalyzed translocation. This conclusion was further confirmed by experiments in which the mean time for tetra-peptide formation $\tau_{Tetra}$ was measured (Fig. 3f). In the absence of THB $\tau_{Tetra}$ was $232 \pm 21$ ms, which increased to $1465 \pm 105$ ms in its presence (Fig. 3f). However, the translocation time $\tau_{Trans}$ estimated using that $\tau_{Tetra} = 3x\ \tau_{Dip} + 2x\ \tau_{Trans}$ (see Supplementary Note 2 for details) showed similar values ($\tau_{Trans} \approx 73 \pm 11$ ms) irrespective of the presence or absence of THB as in case of tripeptide experiments (Fig. 3e). These experiments also demonstrate that in the presence of THB each round of peptide elongation gets prolonged by the same duration (~470 ms) as for the single event of dipeptide formation ($\tau_{Dip}$), which does not include translocation. Thus, we conclude that THB does not affect EF-G dependent translocation. Instead, THB extends the elongation cycle time by impeding aa-tRNA delivery to the ribosome and its accommodation into PTC. Taking also into account that $T_3$ outcompetes THB in the binding to the ribosome (Supplementary Fig. 6) these results indicate that THB does not dissociate /re-associate during each elongation cycle but remains bound to the translating ribosome during the entire period of peptide elongation.

## THB strongly interferes with release factor-dependent peptide release

We next asked whether THB interferes with the binding of class-I RFs in the A site and thereby inhibits stop codon recognition and peptide

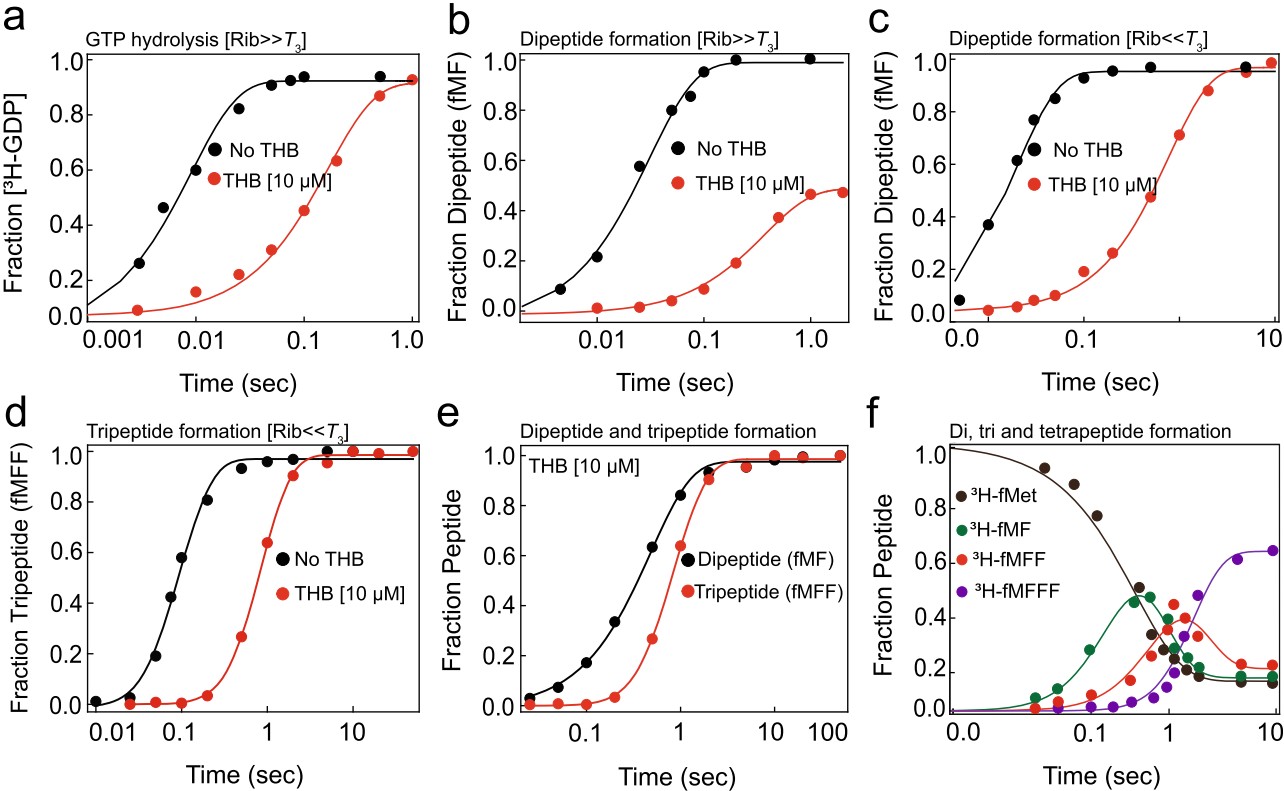

**Fig. 3 | Effects of THB on the elongation cycle of translating ribosomes. a** Time course of GTP hydrolysis on EF-Tu (0.3 µM) in ternary complex ($T_3$) with [³H]GTP and Phe-tRNA^Phe upon its delivery to the MF-mRNA-programmed 70S IC (1 µM) and **b** subsequent dipeptide formation upon tRNA accommodation and peptidyl transfer, without or with THB—pre-incubated with 70S ribosome. **c** Time course of f[³H]Met-Phe dipeptide formation with saturating concentration of $T_3$ (10 µM) in the absence and presence of THB. GTP hydrolysis data were fitted with single-exponential function (Supplementary Eq. 38) and dipeptide data with double-exponential function (Supplementary Eq. 39). **d** Time course of f[³H]Met-Phe-Phe tripeptide formation on MFF mRNA-programmed 70S IC (0.5 µM) mixed with 10 µM $T_3$ and 10 µM EF-G, in the absence and presence of THB. The data were fitted to a three-step sequential model (Supplementary Eq. (52)). **e** Comparison of time courses of di- and tripeptide formation on THB-bound ribosome. **f** Time evolution of dipeptide (f[³H]Met-Phe), tripeptide (f[³H]Met-Phe-Phe) and tetrapeptide (f[³H]Met-Phe-Phe-Phe) upon MFFF mRNA translation by THB-bound 70S ribosomes. The solid lines are fitting of data to a multi-step sequential model (see Supplementary Note 2 for details).

release. To check that, we monitored the kinetics of peptide release by RF2. Here, we used a "minimal" pre-termination complex (Pre-TC) consisting of 70S ribosome programmed with a Met-Stop mRNA and carrying a peptidyl-tRNA analog f[³H]Met-tRNA^fMet in the P-site. The time-course of f[³H]Met release by RF2 was monitored in a quench-flow instrument and the kinetic parameters of the RF2-dependent peptide release reaction was estimated by RF2 titration. The kinetic efficiency ($k_{cat}/K_M$) of peptide release by RF2 in the absence of THB was 6.6 µM$^{-1}$ s$^{-1}$ (Fig. 4a, Supplementary Fig. 13a), with $K_M = 1$ µM and $k_{cat} = 6.5$ s$^{-1}$ (see Supplementary Note 3, Supplementary Eq. (67) for details). The rate of f[³H]Met release at near-saturating RF2 concentration (5 µM) was 5.6 s$^{-1}$, similar to that measured in a previous report[24]. However, when pre-TC was pre-incubated with 5 µM THB, the rate of f[³H]Met release, $k_{obs}$, dropped drastically by 550-fold, from 5.6 s$^{-1}$ to about 0.01 s$^{-1}$ (Fig. 4b). Variation of THB concentration from 0.5 to 10 µM during pre-incubation with pre-TCs had no additional effect on the rate of fMet release (Supplementary Fig. 13b). The rate of f[³H]fMet release in THB's presence could, however, be increased considerably by increasing RF2 concentration (Fig. 4c). The rate $k_{obs}$ increased 2-fold, from 0.01 s$^{-1}$ to about 0.02 s$^{-1}$ when RF2 concentration was increased by 12-fold from 5 to 60 µM RF2 (Fig. 4d). From experiment in Fig. 4d we obtained $K_M$ of ~13 µM and $k_{cat}$ of ~0.02 s$^{-1}$, leading to $k_{cat}/K_M$ estimate of only 0.0015 µM$^{-1}$ s$^{-1}$. This $k_{cat}/K_M$ value is more than three orders of magnitude smaller than that measured in the absence of THB (Fig. 4a). Importantly, the fraction of released f[³H]Met at the end point of the release reaction

was same in the presence and absence of THB. It suggests that even with THB the RF2-dependent peptide release proceeded to completion, albeit significantly slower.

The observed 13-fold increase in the $K_M$ value of RF2-dependent peptide release shows that THB strongly inhibits, but does not abolish completely, the binding of RF2 to the ribosome with the A-site stop codon. However, the huge 4400-fold reduction in the kinetic efficiency, $k_{cat}/K_M$ of the release reaction indicates that THB strongly inhibits the subsequent stop codon recognition and/or the conformational change in RF2 required for proper accommodation of the factor's GGQ motif into the PTC, necessary for peptide release[25,26]. The observed insensitivity of the peptide release rate to increasing THB concentration (Supplementary Fig. 13b) suggests that peptide release can occur on the THB-bound ribosome. However, our kinetic data (Fig. 4, Supplementary Fig. 13b) can also be explained by an alternative model illustrated in Supplementary Fig. 14a in which THB dissociates from pre-TC to allow stop codon recognition by RF2. This alternative model assumes that RF2 can weakly bind to THB containing pre-TC, so that upon dissociation of THB the pre-bound RF2 can immediately recognize the stop codon, change conformation, and release the peptide (see Supplementary Note 3 for detailed discussion of alternative kinetic models of release). According to this 'THB dissociation' model, the rate of peptide release in the presence of THB at saturating RF2 concentration ($k_{cat}$) (see Fig. 4d) should be very close to the THB dissociation rate from pre-TC with pre-bound RF2 which could, therefore, be estimated as 0.02 s$^{-1}$. It means that THB can be

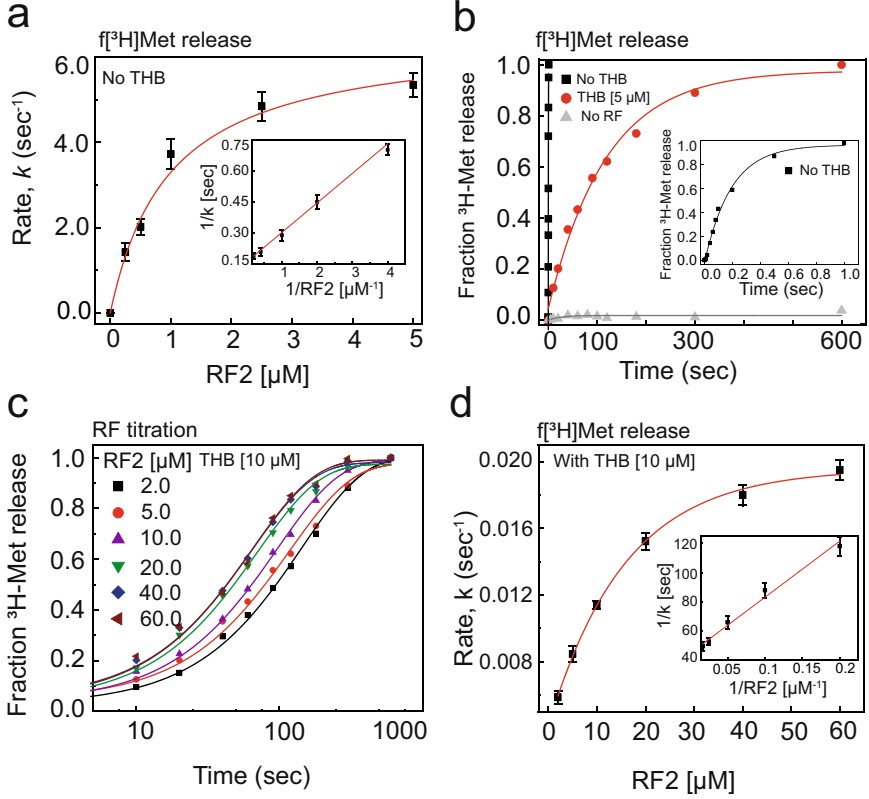

**Fig. 4 | Effects of THB on RF2-mediated peptide release. a** RF2 concentration dependence on the rate of release of f[³H]Met from the pre-TC containing f[³H] Met-RNA in the P site and a UAA stop codon in the A site. Solid line is a hyperbolic fit (Supplementary Eq. 67) of the data. Inset: Double reciprocal plot to estimate the $k_{cat}$ and $K_M$ parameters. **b** Comparison of time courses of f[³H]Met release from pre-TC (0.5 μM) by 5 μM RF2, in the absence and presence of 5 μM THB. Data were fitted with a single-exponential function. **c** Time course of f[³H]Met release from the THB-bound pre-TC (0.5 μM) at various concentrations of RF2. Solid lines indicate single-exponential fit of the data. **d** The dependence of the rate of f[³H]Met release from THB-bound pre-TC with increasing RF2 concentration fitted with hyperbolic function. The $k_{cat}$ and $K_M$ parameters were estimated from the double reciprocal plot (inset). Error bars indicate SEM obtained from independent experiments in triplicates.

dissociated from pre-TC in ~50 s by productive RF2 binding and accommodation, which is much faster than 5 min required for spontaneous dissociation of THB (Supplementary Fig. 9).

### THB prevents ribosome splitting by RRF/EF-G and IF1/IF3 by "tethering" the subunits of the 70S ribosome together

As demonstrated above, THB impedes A-site binding of the $T_3$ and RFs but does not interfere with EF-G binding and subsequent translocation reaction. Splitting of post-termination ribosome complexes (post-TCs) into subunits requires A-site binding of the ribosome recycling factor (RRF) followed by EF-G binding and subsequent GTP hydrolysis on EF-G[27,28]. To study the effect of THB on post-TC splitting reaction we first prepared Met-Phe-Phe mRNA- programmed post-TC carrying deacylated tRNA[Phe] in the P site, and then followed the time-course of the post-TC splitting into subunits in the presence and absence of THB. Briefly, post-TC was rapidly mixed with a factor mix (FM) containing RRF, EF-G, and IF3 in a stopped-flow instrument. The rate of subunit splitting was estimated from the time course of decrease in Rayleigh light scattering at 365 nm[27]. In the absence of THB, post-TC was split into subunits by a concerted action of RRF and EF-G with the rate of $4.7 \pm 0.5 \, s^{-1}$ (Fig. 5a), which agrees well with the previous reports[17,27]. Pre-incubation of post-TC with a small excess of THB inhibited the subunit splitting reaction almost completely (Fig. 5a). Interestingly, in the presence of sub-stoichiometric amounts of THB, only a fraction of post-TC showed subunit splitting. The other fraction, presumably bound with THB, was resistant to RRF/EF-G induced splitting (Supplementary Fig. 15a, b), thus indicating very strong binding of THB to post-TC. Also, increase in RRF concentration, which normally speeds

up the subunit splitting reaction[27], did not promote splitting of the THB-bound post-TC to any tangible degree (Supplementary Fig. 15c).

To check whether THB binding to post-TC would compete with RRF binding, we added THB together with RRF/EF-G/IF3 to the THB-free post-TC in a stopped-flow instrument and monitored post-TC splitting by light scattering. In this experiment we observed that only about a half of post-TC was split into the subunits on a short-time scale (Fig. 5b). Most likely, the other half did bind THB and thereby became resistant to the RRF/EF-G splitting. On a longer time-scale, the subunits generated by fast splitting of THB-free post-TC ribosomes occasionally re-associated, slowly reaching new subunits/70S equilibrium due to stabilization of the re-formed 70S ribosomes by subsequent THB binding (Fig. 5b).

Earlier studies show that the mRNA-programmed tRNA-free ribosome can be efficiently split into subunits by IF1/IF3 added in high concentrations[21]. Here, IF1 binding to the A-site of the ribosome opens the path for the insertion of IF3 in the E-site which leads to the ribosomal splitting into subunits[21]. To check the effect of THB on IF1/IF3 mediated ribosome splitting, we prepared Met-Phe-Phe mRNA-programmed 70S ribosomes and mixed them with IF1 and IF3 in a stopped-flow instrument. Similar to RRF/EF-G mediated splitting of post-TCs, the addition of THB effectively blocked the IF1/IF3-induced splitting of the mRNA-bound 70S ribosomes (Fig. 5c). Furthermore, when THB was added with IF1/IF3 mix, only about half of the ribosomes were split on a fast time-scale and the split subunits re-associated slowly to 70S ribosomes on a longer (~5 min) time scale (Fig. 5c), just like in the case of RRF/EF-G splitting (Fig. 5b). A gradual re-association of the subunits from the already split ribosomes seen in Fig. 5b, c indicates that even in the

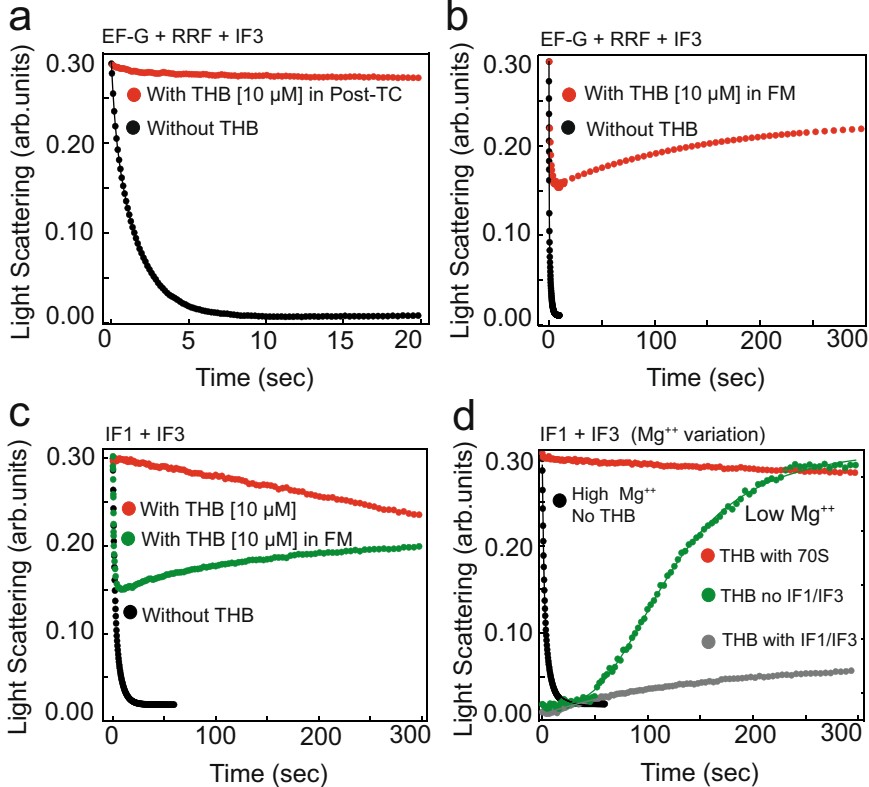

**Fig. 5 | Effects of THB on 70S ribosome splitting and its re-formation from ribosomal subunits. a** Time courses of post-TC (0.25 μM) splitting into subunits by RRF (20 μM), EF-G (10 μM) and IF3 (1 μM). THB (10 μM) was either omitted (black) or added (red) to post-TC. **b** The same as (**a**) but THB (10 μM) was only added with RRF/EF-G/IF3. **c** Time course of splitting of the mRNA-bound 70S into subunits by IF1 and IF3 in the absence (black dots) and presence of THB in the 70S mix (red traces) or in the IF1/IF3 factor mix (FM) (green traces). **d** THB-induced effects on 70S ribosome dissociation (red trace) into and re-formation (green trace) from ribosomal subunits in the presence of 1.3 mM Mg++ in the absence and presence of IF1/IF3 (gray trace). The black trace is the splitting of 70S-mRNA by IF1 and IF3 at 3 mM Mg++ in THB's absence as in '**c**'. Solid lines represent the double-exponential fit of the data and experiments were conducted in triplicates.

presence of IF3, THB can effectively prevent subunit dissociation by binding to the spontaneously re-formed 70S ribosomes and rendering them resistant to IF1/IF3 or RRF/EF-G mediated splitting.

The strong anti-splitting effect of THB could originate from the occlusion of RRF or IF1 binding to the A-site of the ribosome due to THB binding. However, early biochemical results suggested that THB may act as 'glue' that tethers the ribosomal subunits together[10]. To test this, we took advantage of the fact that the ribosomes can be dissociated into subunits by decreasing magnesium (Mg++) concentration[29]. Figure 5d shows that THB-bound 70S ribosome remains resistant to splitting by IF1/IF3 even when the Mg++ concentration was reduced to 1.3 mM. Moreover, when the ribosome was first split into subunits by lowering Mg++ concentration the addition of 10 μM THB shifted the equilibrium towards, albeit slow, formation of 70S ribosome (Fig. 5d). When added together with IF1 and IF3 to the ribosomes split by low Mg++, THB still shifted the equilibrium towards 70S formation (Fig. 5d). The reaction however was very slow due to competition between the factors and THB for binding to the re-associated ribosomes. From these experiments, we conclude that THB, in addition to impeding the A-site binding of RRF and IF1, tethers ribosomal subunits together thereby inhibiting splitting of the post-TC ribosomes and their turnover.

## Cryo-EM structure of THB-bound 70S IC complements kinetic data of THB inhibition

To decipher the structural basis of THB inhibition of different steps of translation as revealed by our kinetic assays, we determined the cryo-EM structure of THB-bound 70S IC (Fig. 6a) comprising *E. coli* 70S ribosome carrying a native-like XR7 mRNA (Met-Phe-Phe-stop), and the

initiator tRNA fMet-tRNA^fMet. The complex was highly homogeneous (~90%) leading to a global resolution of 1.96 Å (Supplementary Fig. 16, Supplementary Table 1). This high resolution allowed us to observe a clear THB density at the intersubunit bridge B2a and a fMet-tRNA^fMet density in the P-site of the ribosome together with mRNA (Fig. 6a–c, Supplementary Fig. 17). The mode of THB binding was similar, but not identical, to that observed in the earlier crystal structure of THB-bound vacant 70S ribosome from *T. thermophilus* (4V8A.PDB) (Supplementary Fig. 18)[13]. The bound THB bridges two major RNA helices of the 30S and 50S subunits; helix h44 of 16S rRNA and helix H69 of 23S rRNA respectively, causing rearrangement of the neighboring RNA nucleotides and forming a number of non-covalent bonds. The tetracyclic moiety of THB stacks between the G1491:C1409 complementary base pair of h44 and A1913 of H69 (Fig. 6c) as also seen in[13]. But the orthohydroxyphenyl group of THB in our structure is in a slightly different orientation (Supplementary Fig. 18). The THB is stabilized by multiple hydrogen bonding - between (i) O9 and O10 of THB with the ribose sugar of A1913; (ii) carboxylic acid moiety of THB with the base of U1915; and (iii) O6 of THB with the ribose sugar of G1491. The nucleobase C1914 of H69 is usually seen stacked with U1915[30,31]. In the presence of THB, the orthohydroxyphenyl group of THB stacks with U1915, flipping C1914 out of the H69 to occupy a region of the vacant A site (Fig. 6c). Both monitoring bases, A1942 and A1943 in h44 are also in their flipped-out conformation, similar to what is seen with aminoglycosides[15]. The high resolution achieved in our cryo-EM map allowed us to unambiguously assign the orientation of the methyl groups of O8 and O9 of bound THB. In addition, it also showed rRNA modifications (both on 16S and 23S rRNAs) and protein side chains conformers (Supplementary Fig. 19) similar to those reported in earlier

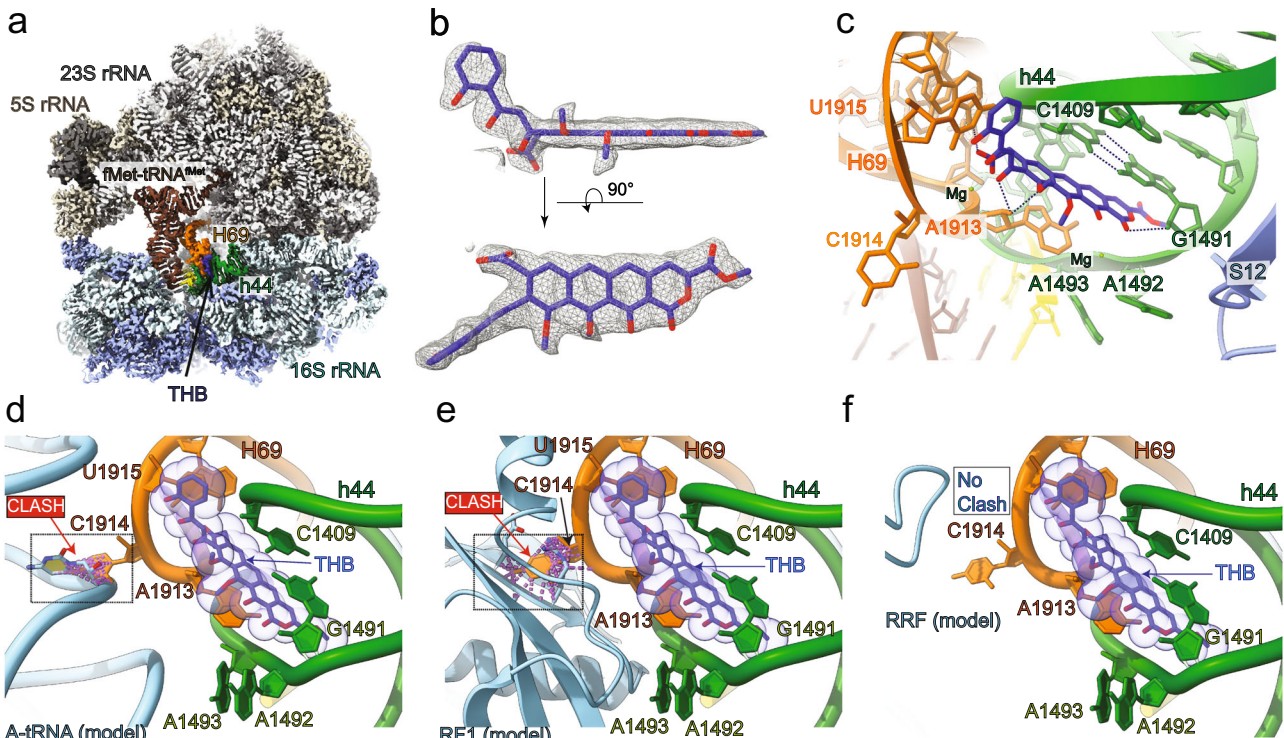

**Fig. 6 | Cryo-EM Structure of THB-bound 70S•mRNA•fMet-tRNAfMet complex.**
**a** Cryo-EM structure of THB-bound *E. coli* 70S IC carrying XR7 mRNA and fMet-tRNAfMet. THB (blue) intercalates between the bases of h44 in the 16S rRNA (green) and H69 of the 23S rRNA (orange) of the mRNA (yellow) bound 70S ribosome. The fMet-tRNAfMet (saddle brown) is stably bound to the P site. **b** EM-density of THB at the binding pocket fitted with the model. **c** Close look at the decoding center reveals interactions of the bound THB with rRNA bases of H69 of 23S rRNA (orange) and h44 of 16 S rRNA (green). Rearrangement of the nucleobases A1913 and C1914 from H69 and A1492 and A1493 from h44 are shown. Dotted lines in black represent hydrogen bonds. **d** THB-induced displacement of C1914 clashes with the accommodated aa-tRNA in the A site as seen from superposition of our 70S-THB structure and aa-tRNA bound 70S ribosome, PDB 7K00[30]. **e** Extensive clashes of extruded base C1914 with multiple amino acid residues (E114, V115, and R116) in the domain II of RF1 from superposition of 70S-THB structure with RF1 bound pre-termination ribosome, PDB 6DNC[47]. **f** THB-induced structural changes at the A site show no sterical clashes with RRF as observed by superposition of 70S-THB structure with RRF bound post-termination ribosome, PDB 4V54[48].

structures of bacterial ribosomes[30,31]. Furthermore, we could observe distinct EM-densities for water molecules coordinating Mg++ ions (Supplementary Fig. 19).

One distinctive feature of our structure is that we observe fMet-tRNAfMet stably bound to the P-site of THB bound 70S ribosome (Fig. 6a). The bound tRNA is fully accommodated into the P-site with its CCA-end contacting the P-loop of 23S rRNA and fMet residue placed in the PTC (Fig. 6b). The anticodon bases of the fMet-tRNAfMet are perfectly Watson-Crick base paired with the AUG codon of the mRNA in the P-site, representing an elongation competent 70S IC, ready to progress to the elongation cycle[32]. In fact, the observed distance between bound THB and fMet-tRNAfMet is more than 20 Å, which explains why THB does not have any influence on P-site binding of the initiator tRNA or its reactivity to puromycin, supporting our biochemical results that THB does not inhibit initiation (Fig. 2).

We further analyze how THB would inhibit elongation, termination and ribosome recycling. A1913 stabilizes the incoming aa-tRNA in the A-site by forming a crucial H-bond with its purine base at position 37[33]. Since A1913 rearranges its position due to multiple stacking interactions with THB, it will no longer be able to make that H-bond. In addition, the flipped-out base C1914 will clash with the incoming aa-tRNA, as seen from overlay of the aa-tRNA bound ribosome structure (PDB 7K00) (Fig. 6d). This explains how THB inhibits aa-tRNA delivery and accommodation. When compared with RF1 bound ribosome structure (PDB 6DNC) we find that the displaced C1914 would create severe clashes with multiple amino acid residues in the domain II of class I RFs (Fig. 6e). Thus, most likely THB binding will strongly inhibit RF1/RF2 binding and accommodation, thereby inhibiting termination.

However, both in case of aa-tRNA and RFs, further conformational rearrangement of the rRNA bases might allow their binding, although in a highly defective manner.

When compared with EF-G and RRF/EF-G bound structures with our THB bound ribosome structure (PDB 4V54), no obvious molecular clashes were found (Fig. 6f and Supplementary Fig. 20). This analysis provides structural explanation for why no inhibition by THB was observed for EF-G mediated translocation. It also indicates that THB's inhibition of RRF/EF-G and IF1/IF3 mediated splitting could hardly be explained by impediment of RRF and IF1 binding in the A site.

## Discussion

THB has been historically classified as the inhibitor of protein synthesis initiation that prevents the initiator tRNA from binding to mRNA-programmed bacterial ribosomes[11]. This classification was based on biochemical experiments in which random poly-(A,U,G) hetero-polymer was used instead of native mRNA[10–12]. The results of our translation initiation assays (Fig. 2) and our 1.96 Å resolution cryo-EM structure of a THB-bound functional *E. coli* 70S initiation complex carrying proper mRNA and fMet-tRNAfMet (Fig. 6) are, however, at odds with the THB classification as the inhibitor of translation initiation. The co-occurrence of THB and the P-site bound fMet-tRNAfMet on the mRNA-programmed ribosome in our structure clearly demonstrates that THB does not preclude fMet-tRNAfMet binding to the P site. This is also confirmed by our nitrocellulose filter-binding assays with f[³H]Met-tRNAfMet (Fig. 2c) and real-time binding of BOP•Met-tRNAfMet to the 70S ribosomes (Fig. 2d); in both cases we used native-like mRNA (containing SD sequence and AUG start codon followed by other codons). We show

that THB does not diminish f[³H]Met-tRNA$^{fMet}$ binding even when the ribosomes are pre-incubated with it (Fig. 2b). Moreover, the 70S-bound f[³H]Met-tRNA$^{fMet}$ remains puromycin-reactive in the presence of THB (Supplementary Fig. 3); that reconfirms its P-site location. Furthermore, THB does not inhibit subunit association irrespective of the presence or absence of the IFs (Fig. 2a, Supplementary Fig. 2), suggesting that THB binds weakly to the individual subunits and more importantly that THB does not inhibit any step of translation initiation.

When the subunits are associated forming the 70S ribosome THB binds very strongly[10] at the intersubunit bridge B2a near the decoding center (DC) involving h44 and H69 (Fig. 6). It suggests that THB likely interferes with the subsequent steps of protein synthesis that involve A-site binding of aa-tRNAs and the translation factors. Indeed, our experiments show that THB causes an about 20-fold reduction in the rate of aa-tRNA delivery to the A-site of the ribosome by EF-Tu•GTP in $T_3$ (Fig. 3a) accompanied by an additional minimum 25-fold reduction in the rate of subsequent aa-tRNA accommodation into the PTC, thereby leading to ~0.5 s delay at each round of peptide elongation (Fig. 3b). The THB interference with tRNA accommodation results also in a high drop-off frequency of the delivered aa-tRNA from the A-site (Fig. 3b). The two kinetic effects of THB on dipeptide formation are well explained by our cryo-EM structure (Fig. 6), in which THB binding to the intersubunit bridge B2a causes repositioning of A1913 and an orthogonal rotation of C1914, two important nucleobases at the tip of H69, as reported earlier for *T. thermophilus* ribosome[13]. This structural remodeling of nucleobases in the A-site sterically interferes with $T_3$ binding to the decoding center of the 30S subunit (Fig. 6d) and subsequent aa-tRNA accommodation, explaining both impaired binding and increased aa-tRNA drop-off in the presence of THB. The repositioned C1914 is probably in a dynamic state since $T_3$ binding and aa-tRNA accommodation do eventually occur (Figs. 3a and 3b). It should be noted, however, that the drop-off of aa-tRNA in the presence of THB is not expected to diminish the processivity of translation in vivo since the dropped off aa-tRNA can be re-delivered to the A-site by another EF-Tu•GTP (Fig. 3c). Our results thus demonstrate that THB impedes aa-tRNA delivery and their accommodation at the A site.

We observed an approximately equal (20-fold) reduction of the efficiency of aa-tRNA delivery for both cognate and near-cognate codons in the presence of THB (Fig. 3a, Supplementary Fig. 11). It implies that THB, despite flipping-out the monitoring bases A1492 and A1493, does not affect the accuracy of aa-tRNA selection. It also means that contrary to aminoglycosides[17,34] and tuberactinomycins[35] that primarily act by flipping-out the monitoring bases, THB does not stabilize aa-tRNA in the A-site. Rather, the repositioned C1914 in the A-site acts as the main obstruction for stable and rapid binding of aa-tRNA and induces drop off of both cognate and near-cognate tRNA prior to A-site accommodation.

We have studied the effect of THB on EF-G dependent translocation by monitoring the mean times of tri- and tetra-peptide formation (Fig. 3f). The mean translocation time ($\tau_{Trans}$) is very similar (~60 ms) in the presence and absence of THB, despite drastically different mean time of tripeptide formation ($\tau_{Trip}$) of ~100 ms in the absence and ~1 s in the presence of THB (Fig. 3d). A similar calculation for tetrapeptide formation also showed that the actual translocation time is not affected by THB. Thus, our results strongly suggest that THB does not inhibit EF-G mediated translocation. The observed delaying effects of THB on tri- and tetra-peptide formation stem, instead, from ~0.5 s delay at every elongation round due to prolongation of aa-tRNA delivery and accommodation (as discussed above). This explanation is further strengthened by the observed insensitivity of $\tau_{Trip}$ to EF-G concentration in the presence of THB, which means that the translocation at a lower EF-G concentration is already much faster than aa-tRNA delivery/accommodation to THB-bound ribosome. The insensitivity of EF-G catalyzed translocation to THB could also be explained from structural argument. In order to allow accommodation of aa- or

peptidyl tRNA in the A-site the nucleobase C1914 must be deflected back towards its in-helix position, which is a pre-requisite for the binding of EF-G and subsequent translocation[36,37]. Consistent with this, our structural analysis does not show any steric clash of rRNA bases in THB-bound state with EF-G (Supplementary Fig. 20).

Our results also exclude the possibility of THB dissociation before or during translocation, since, otherwise, the formation of subsequent peptide bonds in our tri- and tetra-peptide experiment should occur fast because of much slower binding of THB than $T_3$ to the ribosome (Supplementary Fig. 6). Our tri- and tetra-peptide formation kinetics show, on the contrary, that the formation of the second and third peptide bonds on THB pre-incubated ribosomes is delayed by the same 0.5 s as for the first peptide bond (Fig. 3d, f). Thus, we conclude that EF-G mediated translocation occurs uninhibited despite THB remaining bound to the elongating ribosome. It also suggests that THB has no drastic effect on the ribosomal intersubunit dynamics during translocation.

Despite a large decrease in the rate of aa-tRNA delivery/accommodation, THB exerts its most striking effects by inhibiting translation termination and subsequent ribosome recycling. We found that even at very high RF2 concentrations THB delays RF2-dependent peptide release by ~50 s (Fig. 4d). The peptide release at the physiological RF concentration (≈2 μM) will, therefore, presumably take well above 45 s, causing major halt in protein production in the bacterial cell. Our results suggest that THB interferes severely with both RF1/RF2 binding to the A-site and their conformational change for accommodation to PTC required for peptide release[25,38]. These effects most likely originate from THB-induced extrusion of 23S rRNA base C1914 that generates extensive clashes with multiple amino acid residues of the domain II of A-site bound RF1/RF2 (Fig. 6e). Moreover, THB-induced flipping out of the monitoring bases A1492 and A1493, and repositioning of A1913 are also incompatible with the binding of class I RFs[39]. These structural obstructions are much more severe than that for A-site bound aa-tRNA (Fig. 6d). Thus, most likely THB needs to dissociate to allow productive binding and peptide release by RF1/RF2. This can be seen from our kinetics data that a pre-bound RF can destabilize THB and hasten THB dissociation in ~50 s (Fig. 4d) which would otherwise take 5 min (Supplementary Fig. 9). However, the possibility of an alternative model where peptide release can occur very slowly by conformational rearrangements on THB-bound ribosome cannot be ruled out.

We have found that THB very effectively prevents the RRF/EF-G or IF1/IF3 mediated splitting of THB-bound post-TC or tRNA-free mRNA-bound 70S, respectively, into the ribosomal subunits (Fig. 5a–c). Inhibition of RRF/EF-G mediated splitting can partially be explained by the incompatibility of RRF binding to the classical/non-rotated state of the ribosome stabilized by THB[28]. However, this explanation alone cannot account for strong anti-dissociation effect of THB observed at low Mg$^{++}$ concentration with and without splitting factors (Fig. 5d). Our cryo-EM structure clearly positions THB at the interface of H69 and h44, where it acts as a bridge between 50S and 30S thereby tethering the subunits like a molecular-glue. We propose therefore that it is the subunit gluing, or tethering effect of THB that makes it such a powerful inhibitor of subunit dissociation. The THB inhibition of ribosome splitting would halt ribosomes at the recycling stage and inhibit the subsequent initiation events[21], thereby blocking the protein synthesis in the bacterial cell.

Recent studies of the molecular mechanisms of action of the ribosome-targeting antibiotics have revealed that many of them inhibit bacterial protein synthesis by affecting not only one, but several steps of the ribosomal translation cycle. For instance, aminoglycoside antibiotics known to reduce the accuracy of tRNA selection were recently found to also affect the translocation step of the protein elongation cycle as well as termination[17,40]. Our kinetic and structural studies of THB revealed that this drug, like many other antibiotics, targets multiple stages of translation. However, despite binding to the ribosomes at the site close to that of aminoglycosides[17] and tuberactinomycins[14]

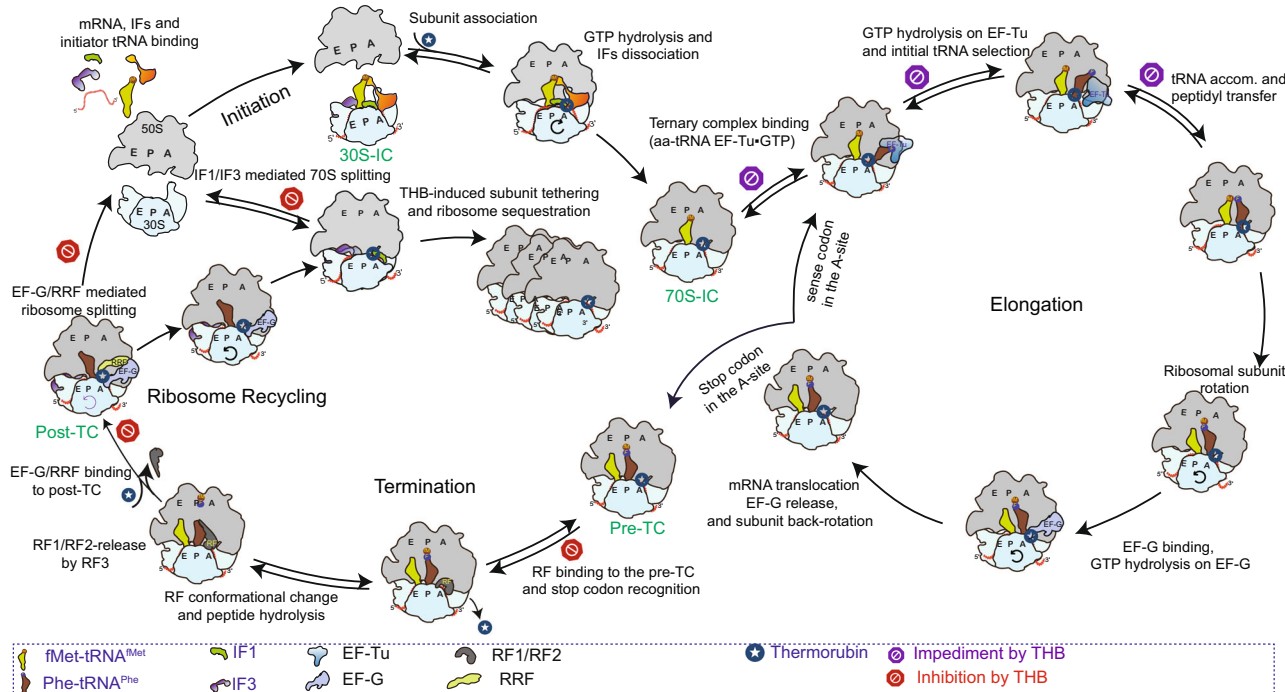

**Fig. 7 | Overall mechanism of THB inhibition of bacterial translation cycle.** THB binds tightly to the 70S IC upon subunit joining but has no effect on initiator tRNA binding to the 30S pre-IC and subsequent 70S IC formation. THB impedes EF-Tu•GTP mediated delivery of the aa-tRNA to the A site and its subsequent accommodation into PTC during translation elongation. However, THB permits ribosomal subunit rotation, EF-G binding and translocation of peptidyl-tRNA from the A- to the P-site. During termination, THB inhibits the binding of RF1/RF2 to the stop codon at the A site of pre-TC and subsequent conformational change in the RFs required for peptide release. Further, THB prevents the splitting of post-TC into subunits by RRF and EF-G. This effect primarily originates from the 'gluing' effect of THB, which tethers the subunits together. This effect also causes re-association of 70S from already dissociated subunits in the absence of IF3. The red signs indicate the steps THB inhibits during translation cycle, while the purple signs indicate the steps impeded by THB. All other steps including initiation and translocation remain unaffected by THB.

(Supplementary Fig. 21), it targets the elongation, termination, and ribosome recycling steps in a unique and rather unanticipated manner. In contrast to the aminoglycosides, which stabilize both cognate and near/non cognate aa-tRNAs at the A site and thereby compromise accuracy of tRNA selection, THB severely impedes delivery and accommodation of the aa-tRNAs at the A-site, without affecting the accuracy of tRNA selection. In this regard, the action of THB is reminiscent to that of the structurally similar antibiotic tetracycline, although the binding site of the latter differs from that of THB[14,15]. Also, while aminoglycosides slow down the EF-G mediated translocation step of the elongation cycle[17,41], THB seems to have no effect on it despite binding to the bridge B2a and flipping-out the monitoring bases A1492 and A1493 similar to the aminoglycosides. Thus, THB binding probably allows more conformational dynamics of A1492 and A1493, while aminoglycosides do not.

In summary, we present here a complete characterization of the mechanisms of inhibition of bacterial protein synthesis by THB (Fig. 7). First of all, our results refute the earlier claim[11] that THB inhibits the initiation step of translation by preventing initiator tRNA binding to the P site. Secondly, although THB causes significant delay in aa-tRNA delivery to the A-site and its accommodation into PTC, our results reveal that THB exerts its strongest deleterious effect on bacterial protein synthesis by inhibiting RF-mediated peptide release and subsequent ribosome recycling, thereby sequestering translating ribosomes in pre- and post-termination states. However, limited solubility of THB prevents it from being widely used clinically as a microbial inhibitor[8]. The rational modifications of THB aimed at increasing its solubility could make it an attractive therapeutic option against multiple drug-resistant infections. Moreover, the unique binding site of THB could be used for rational design of novel antibiotics to meet the current global needs for effective antibiotics.

## Methods

### Buffers and reagents

All in vitro translation experiments were conducted at 37 °C in HEPES–Polymix buffer (pH 7.5) (5 mM HEPES (pH 7.5), 95 mM KCl, 5 mM NH$_4$Cl, 5 mM Mg(OAc)$_2$, 8 mM putrescine, 0.5 mM CaCl$_2$, 1 mM spermidine and 1 mM 1,4-dithioerythritol) with additional energy regeneration components including 10 mM phosphoenolpyruvate (PEP), 50 μg/ml pyruvate kinase (PK), 2 μg/ml myokinase (MK), and 1 mM of ATP and GTP (unless otherwise specified). *E. coli* MRE600 cells were used to prepare tight-coupled 70S ribosomes following standard laboratory protocol. His-tagged clones of the translation factors (IF1, IF2, IF3, EF-Tu, EF-Ts, EF-G, RF1, RF2, RF3, and RRF) and Phenylalanine tRNA synthetase were overexpressed in *E. coli* BL21 (DE3) cells and purified using nickel-affinity chromatography (HiTrap; GE Healthcare). f[$^3$H]Met-tRNA$^{fMet}$ and tRNA$^{Phe}$ were prepared as described[17]. BODIPY™ (BOP)•Met-tRNA$^{fMet}$ was prepared as described earlier[38]. In vitro transcribed XR7-mRNAs with strong Shine-Dalgarno sequence (AAGGAGG) and a small ORF AUGUUCUUCUAA (Met-Phe-Phe-stop), AUGCUCUU-CUAA (Met-Leu-Phe-stop), AUGUUCUUCUUCUAA (Met-Phe-Phe-Phe-stop), and AUGUAA (Met-stop) were prepared according to Holm et al.[35] Concentrations of the ribosomes, translation factors, tRNAs, and mRNAs were measured spectrophotometrically. THB (Thermorubin A) was purchased from AdipoGen Life Sciences and was dissolved in Dimethyl sulfoxide (DMSO) solution and stored at −20 °C. [$^3$H] Met and [$^3$H] GTP were from Perkin Elmer and all other chemicals were from either Merck or Sigma-Aldrich.

### Synthesis of a reporter protein GFP+ in a fully reconstituted *E. coli* translation system

We used our reconstituted cell-free translation system containing active 70S ribosomes (1 μM), translation factors (1–10 μM), 20

aminoacyl tRNA synthetases (1–10 U), and tRNAs (bulk tRNA 100 μM) purified from *E. coli* along with 20 amino acids and energy pump components[19]. The reaction was started by the addition of in vitro transcribed and purified mRNA encoding a reporter protein (GFP+). To test the effect of THB on GFP+synthesis, we added THB in various concentrations (0–1 μM) to the reaction mixture. The synthesis of GFP+ (at 37 °C) was monitored in real-time as the time-dependent increase in GFP+ fluorescence signal (excitation – 460 nm and emission – 518 nm) using a TECAN Infinite 200 PRO multimode plate reader. The experiments were done in triplicates.

### Subunit association assay

Two reaction mixes were prepared and incubated separately at 37 °C for 10 min. Mix A – 30S ribosome (0.5 μM), mRNA (Met-Phe-stop) (2 μM), IF1 (2 μM), IF2 (2 μM), IF3 (1 μM), and fMet-tRNA$^{fMet}$ (2 μM) and Mix B – 50S ribosome (0.5 μM). The reaction was started by rapid mixing of the Mixes A and B in a stopped-flow instrument (SX-20, Applied Photophysics) equilibrated at 37 °C. All IFs were omitted for naked subunit association. The time-course of subunit association was monitored by following the increase in Rayleigh light scattering at 365 nm[20]. For THB experiments, 10 μM of THB was added to both mixes. The average data (from three to five individual experiments) were fitted with complex formation function (Supplementary Eq. (9) in Supplementary Note 1) using OriginPro 2016 (OriginLab Corp).

### BOP•Met-tRNA$^{fMet}$ binding to mRNA-programmed 70S ribosomes

Two mixes were prepared and incubated separately at 37 °C for 10 min. Mix A – 70S (0.25 μM), mRNA (Met-Phe-Phe-stop) (2 μM), IF1 (1 μM), IF2 (1 μM), and IF3 (0.5 μM) and Mix B - BOP•Met-tRNA$^{fMet}$ (0.3 μM). THB (10 μM) was added to both mixes. Equal volumes of the two mixes were rapidly mixed in a stopped-flow instrument (μSFM, BioLogic, France) at 37 °C and the resultant increase in BOP fluorescence (at 590 nm) due to the binding of BOP•Met-tRNA$^{fMet}$ to the 70S ribosomes as a function of time was monitored. For other experiments, either IF2 was incubated with BOP•Met-tRNA$^{fMet}$ in Mix B, or all IFs were absent. Experiments were carried out in triplicates and average data were fitted with an exponential function (Supplementary Eq. (8) in Supplementary Note 1) using OriginPro 2016 (Origin LabCorp).

### Nitrocellulose filter-binding assay to monitor binding of f[³H]Met-tRNA$^{fMet}$ binding to mRNA-programmed 70S ribosomes

The nitrocellulose filter-binding assay was performed to measure the extent of initiator tRNA (f[3H]Met-tRNA$^{fMet}$) binding to the mRNA-programmed 70S ribosomes[21]. In this assay, 70S ribosomes (0.5 μM) were incubated with mRNA (Met-Phe-Phe-stop) (2 μM), IFs (1 μM each), and f[3H]Met-tRNA$^{fMet}$ (0.6 μM) for 15 min at 37 °C to prepare 70S IC. THB (10 μM) was added to the IM before or after incubation (as indicated). To check f[3H]Met-tRNA$^{fMet}$ binding, an aliquot (50 μL) of 70S IC was applied to a nitrocellulose filter membrane (pre-soaked in HEPES–polymix buffer (pH 7.5)) connected to an air suction filtration assembly and washed twice with ice-cold HEPES–polymix buffer (5 mL). The radioactivity [³H] retained in the filter (representing the bound f[3H]Met-tRNA$^{fMet}$ to 70S ribosomes) was measured using QuickSafe scintillation cocktail in Beckman LC6500 counter. The experiments were repeated at least three times and the average radioactivity count (cpm) was plotted for each case.

### Puromycin assay to verify fMet-tRNA$^{fMet}$ accommodation in the P site and its activity in peptidyl transfer

To check the accommodation of the initiator tRNA in the P-site of the 70S ribosome and subsequent activity in peptide bond formation we performed the puromycin (Pmn) assay. For that, we prepared an initiation mix (IM) by incubating 70S ribosomes (0.5 μM), mRNA (Met-Phe-Phe-stop) (1 μM), IF1(1 μM), IF2 (1 μM), IF3 (0.5 μM), and f[³H]-Met-

tRNA$^{fMet}$ (0.6 μM) at 37 °C for 15 min. Next, 200 μM of Pmn was added and incubated for another 5 min. THB (10 μM) was either pre-incubated with IM or added later to IM. The reaction was quenched by adding 50% formic acid and the samples were centrifuged at 14000 rpm for 15 min. Supernatant and pellet were separated, and 0.5 M KOH (165 μL) was added to the pellet. To both supernatant and pellet, 33% methanol was added and centrifuged at 14000 rpm for 15 min to extract [³H]fMet-Pmn formed. Radioactive [³H] counts (cpm) in both supernatant and pellets were measured using QuickSafe scintillation cocktail in Beckman LC6500 counter. Experiments were performed in triplicates along with negative controls (no Pmn, no THB, or no 70S).

### Evaluation of association and dissociation rate constants of THB

To measure the ribosome binding kinetics of THB, 70S ribosomes (1 or 2 μM) in HEPES–polymix buffer were rapidly mixed with 1 μM THB in a stopped-flow instrument (SX-20, Applied Photophysics) at 37 °C. The resultant increase in fluorescence (at 340 nm) due to the binding of THB to the 70S ribosome was monitored over time. The fluorescence traces obtained in the reactions were fitted with complex formation equations (Supplementary Eqs. 8 and 9). To estimate dissociation rate of THB from the ribosome, we prepared THB-70S complex or THB 70S-IC by incubating THB (1 μM) with 70S/70S IC (2 μM) at 37 °C for 5 min. This complex was mixed with a large molar excess (50 μM) of aminoglycoside arbekacin (ABK) in the stopped-flow instrument and resultant decrease in fluorescence (at 340 nm) as a result of THB dissociation was monitored. The fluorescence traces were fitted to an exponential function (Supplementary Eq. 19) using OriginPro 2016 (Origin LabCorp) and rates of THB dissociation were estimated. Experiments were conducted in duplicates.

### GTP hydrolysis on EF-Tu, dipeptide, tripeptide, and tetrapeptide formation experiments

GTP hydrolysis and dipeptide formation experiments were conducted as described earlier[17]. For di-, tri-, and tetra-peptide formation experiments, initiation mix (IM) and elongation mix (EM) were prepared by incubating the following components in their final concentrations as indicated. IM – 70S (0.5 μM), mRNA (Met-Phe-Phe-stop/Met-Phe-Phe-Phe-stop) (1.5 μM), IF1 (1 μM), IF2 (0.75 μM), IF3 (1 μM), f[3H]Met-tRNA$^{fMet}$ (0.6 μM), 1 mM GTP and 1 mM ATP. EM - EF-Tu (10 μM), Phe-tRNA$^{Phe}$ (2-12 μM), Phenylalanine (200 μM), Phe-RS (0.5 μM), GTP (1 mM) and ATP (1 mM). In tri-and tetra-peptide experiments, EM was supplemented with EF-G (10 μM). THB was added to IM or EM or to both as indicated. Both IM and EM were incubated at 37 °C for 15 min. Equal volumes of each mix were rapidly mixed in a quench-flow instrument (RQF-3; KinTek, Corp.) and the reaction was quenched with 17% formic acid at different incubation intervals. Near-cognate reactions of GTP hydrolysis and dipeptide experiments were slow; therefore, the samples were mixed manually. The samples were processed as described earlier[17]. The average times of GTP hydrolysis ($\tau_{GTP}$), dipeptide ($\tau_{Dip}$), tRNA accommodation-peptidyl transfer ($\tau_{A/PT}$), tripeptide ($\tau_{Trip}$), and tetrapeptide ($\tau_{Tetra}$) were estimated as described in Supplementary Note 2. Experiments were conducted in triplicates together with control experiments.

### Experiments for release factor-mediated peptide release

Pre-termination complexes (pre-TCs) (70S programmed with mRNA(Met-stop) carrying f[³H]fMet-tRNA$^{fMet}$ in the P site and a stop codon (UAA) in A site), were prepared by mixing together 70S ribosome (2 μM), IF1 (2 μM), IF2 (2 μM), IF3 (2 μM), XR7 mRNA (10 μM), f[³H]Met-tRNA$^{fMet}$ (2.5 μM). Later, 2 mM Mg(OAc)$_2$ was added to the mixture for stabilization of the complex. After incubation at 37 °C for 20 min the pre-TC was purified by ultracentrifugation (259,000 × *g*) through a sucrose cushion (1.1 M sucrose) containing an additional 4 mM of Mg(OAc)$_2$ at 4 °C for 2 h. The pellet containing pre-TC was resuspended in HEPES–polymix Buffer (pH 7.5) and stored at −80 °C.

For peptide release experiments, equal volumes of pre-TC (0.5 μM) and release factor (RF2) (0.5–10 μM), were rapidly mixed at 37 °C in a temperature-controlled quench-flow (QF) instrument (RQF-3; KinTek, Corp.) and samples were quenched at defined time intervals using HCOOH to a final concentration of 17%. The samples were centrifuged in the cold room (4 °C) at 20,000 × *g* for 15 min to separate released f[3H]Met in the supernatant and intact f[3H]Met-tRNA in the pellet. The pellets were treated with 165 μL of 0.5 M KOH at room temperature for 10 min to release f[3H]Met. The amount of radioactive f[3H]Met in supernatant and pellet was determined using QuickSafe Flow scintillation cocktail in Beckman LS6500 counter. As controls, aliquots of pre-TC were taken before and after QF experiments to monitor the spontaneous hydrolysis of f[3H]Met from f[3H]Met-tRNA^fMet in pre-TC by manual quenching. Experiments were performed in triplicates and average data were analyzed using the single-exponential function in OriginPro 2016 (Origin LabCorp).

### Experiments for RRF/EF-G and IF1/IF3 mediated ribosome recycling and subunit re-association

The post-termination ribosome complex (post-TC), with an empty A-site and deacylated tRNA in the P-site, was prepared by incubating 70S ribosome (0.5 μM) with mRNA (Met-Phe-Phe-stop) (1 μM) and deacylated tRNA^Phe (1 μM) for 5 min at 37 °C. Similarly, a factor mix (FM) containing RRF (20 μM), EF-G (10 μM), and IF3 (1 μM) was prepared. THB (10 μM) was either incubated with post-TC or added to the FM mix as indicated. Equal volumes of post-TC and FM were rapidly mixed in a stopped-flow instrument (SX-20, Applied Photophysics) and the time course of post-TC splitting into subunits was monitored as the decrease in Rayleigh light scattering at 365 nm[27]. The rate of post-TC splitting was estimated by fitting the data with the double-exponential equation in OriginPro 2016 (Origin LabCorp). Similarly, RRF (10–60 μM) was titrated for concentration-dependent effects against THB bound post-TC. Experiments were carried out in triplicates.

Alternatively, for IF1/IF3 mediated splitting assay[21], 70S ribosomes (0.5 μM) were incubated just with mRNA (Met-Phe-Phe-stop) (1 μM) for 15 min at 37 °C. Likewise, an initiation factor (IF) mix containing IF1 (8 μM) and IF3 (2 μM) was prepared. THB was added in the mRNA-70S mix or IF mix as indicated. The kinetics of dissociation of tRNA-free 70S into ribosomal subunits was monitored as described above. Moreover, for the assessment of THB-induced re-association of subunits, the mRNA-70S complex (0.5 μM) was first split into subunits by lowering the concentration of Mg++ to 1.3 mM (by increasing the concentration of Mg++ chelators; from 1 mM of ATP to 2 mM). IF mix either contained both IF1 (8 μM) and IF3 (2 μM) or none. Both mixes were incubated for 10 min at 37 °C and rapidly mixed in a stopped-flow instrument. The time-course of Rayleigh light scattering at 365 nm was fitted to the exponential function as above.

### Cryo-EM sample preparation and data collection

We first incubated 70S ribosomes (200 nM) with THB (10 μM) for 5 min at 37 °C. To this, XR7 mRNA (Met-Phe-Phe-Stop) (2 μM) and fMet-tRNA^fMet (5 μM) were added and the mixture was incubated for another 10 min. Quantifoil grids (R2/2 continuous carbon, 300 mesh, copper) were glow discharged (20 mA, 30 s, PELCO easiGlow), and 3 μL of the sample (mixture) was added onto a grid, the excess sample was blotted off (blot force: 0 and blot time: 3 s) and vitrification of the sample was performed by plunging the grid into liquid ethane using a Mark IV Vitrobot (FEI, ThermoFisher) operated at 4 °C and 95 % humidity. Grids were screened on a Glacios cryo-electron microscope (at 200 keV) with falcon III detector (at the cryo-EM facility, Uppsala). The final data collection was carried out on a Titan-Krios (FEI, ThermoFisher) (SciLife Lab, Stockholm, Sweden) at 300 keV using a K3 direct electron detector (Gatan) with defocus range of −0.6 to −2.0 μm and electron dose of 1.025 e/Å² per frame. A nominal magnification of 105,000x was applied, resulting in a final object sampling resolution of 0.8617 Å/pixel. A total of 13,944 micrographs were collected over 24 h.

### Data processing, model building, and structure determination

Data processing was carried out using comprehensive data processing package in CryoSPARC v3.3.1[42]. After patch motion correction and contrast transfer function (CTF) estimation, a total of 13,914 micrographs were selected for particle picking. Using blob-picker and followed by 2D-template-based particle picking, we extracted a total of 1,930,095 particles. This was followed by several rounds of 2D classification of the particles and selection of good particles while discarding junk particles. We curated a total of 1,776,568 particles from 2D classification and used them for the ab-initio reconstruction of 3D volumes. A total of 1,132,682 particles from good 3D volumes (after removing preferentially oriented and anisotropic particles) were selected and further refined using heterogenous 3D refinement, homogenous refinement (legacy), and finally homogenous 3D refinement. A final map of our complex at 2.03 Å resolution was again processed to global and local CTF refinement, adjusted for per-particle defocus, and sharpened using the map sharpening function in CryoSPARC. The cryo-EM maps were visualized at each stage of 3D refinement using USCF chimera[43]. Details of the cryo-EM data processing are presented in Supplementary Fig. 16.

The structure of the *E. coli* 70S ribosome (PDB ID 7K00)[30] was used as a template for model building. First, model-to-map alignment was carried out using UCSF Chimera[43]. The map of THB-70S•mRNA•f-Met-tRNA^fMet (THB-70S IC) complex was refined using the Phenix real-space refinement[44,45]. The protein and rRNA chains were visually checked in Coot[46] and manually adjusted wherever necessary. Antibiotic THB was manually docked into the experimental density, followed by real-space refinement in Phenix[44]. All figures showing atomic models were generated using UCSF Chimera[43] and labeled with Adobe Illustrator (Adobe Inc.).

### Reporting summary

Further information on research design is available in the Nature Portfolio Reporting Summary linked to this article.

## Data availability

The data that support this study are present in main text or in the supplementary data. The atomic coordinates of the 70S-THB complex has been deposited in the PDB under the accession code 8AYE [https://doi.org/10.2210/pdb8AYE/pdb]. The electron density map generated in this study has been deposited in the Electron Microscopy Data Bank (EMDB) with accession code EMDB-15712. Source data are provided with this paper.

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

## Acknowledgements

The authors thank Raymond Fowler for purification of the translation factors. We would also like to thank Dr. Daniel Larsson and Dr. Anna Sunborger-Lunna, cryo-EM facility Uppsala, for their technical support in grid screening and guidance for data processing. We would like to thank Cryo-EM Swedish National Facility, funded by the Knut and Alice Wallenberg, Family Erling Persson and Kempe Foundations, SciLifeLab, Stockholm University for cryo-EM data collection. This work was supported by research grants from Swedish Research Council [2016-06264, 2018-05946, 2018-05498]; Knut and Alice Wallenberg Foundation [KAW 2017.0055]; Carl Trygger's Foundation [CTS 18:338, CTS 19:806];

Wenner-Gren Foundation [UPD2017:0238] to S.S. N.P.P. was supported by a doctoral scholarship from Sven och Lilly Lawskis fond för Naturvetenskaplig Forskning. Funding for open access: Uppsala University.

## Author contributions

N.P.P.: Project idea, all biochemical experiments (except GFP⁺ synthesis), data analysis, cryo-EM sample preparation, data processing and structural analysis. A.E.: Cryo-EM data processing, model building and structural analysis. C.S.M.: GFP⁺ synthesis assay and data analysis. M.Y.P.: Biochemical data analysis and supervision. S.S.: Funding acquisition, supervision, and data analysis. All authors contributed to manuscript preparation.

## Funding

## Competing interests

The authors declare no competing interests.
