## [Peer Review File · Nature Communications]

Antibiotic Thermorubin Tethers Ribosomal Subunits and Impedes A-site Interactions to Perturb Protein Synthesis in BacteriaREVIEWER COMMENTS

Reviewer #1 (Remarks to the Author):

The paper of Parajuli et al. explores the mechanism of action of the ribosome-targeting antibiotic thermorubin, a long-known drug but with an unclear mechanism of action. By running an array of kinetic assays and obtaining a high-resolution structure of the antibiotic bound to the ribosomal functional complex, the authors demonstrate that the drug interferes with accommodation of aminoacyl-tRNA in the ribosomal A site and with the action of class I release factors and ribosome recycling factor. As the result, in contrast to the previous assertion that thermorubin inhibits initiation of translation, the authors conclude that the drug in fact interferes with translation elongation, termination and recycling.

Critique

It is an interesting and well performed study. The experiments are well designed and well described, the results are convincing, and the conclusions are reasonable. The paper is well written and is an easy read.

My only main problem is that discussing the structure of the ribosome-antibiotic complex, the authors do not analyze possible clash of thermorubin with the A-site tRNA, RF or RRF. On p.16. (ll. 7-8) they allude that bases C1914 and U1915 'occupy the region of the vacant A site'. Do they imply that there is a direct clash with "aminoacyl-tRNA, class-I RFs, and RRF"? If so, how bad is the clash? If not, how do they explain their data? Any of these scenarios need to be properly illustrated using the available relevant structures and discussed in specific terms rather than using vague general statements.

The other problems are minor:

p.8, top. besides three steps of the elongation cycle mentioned by the authors, it also includes peptide bond formation.

Fig. 3E. correct typo in the text line above the graph

p.12, bottom. It was unclear to me where the conclusion that THB needs to dissociate from the ribosome for peptidyl-tRNA hydrolysis to take place came from. What data show that RF cannot promote peptidyl-tRNA hydrolysis on the THB-bound ribosome, albeit slowly?

p.14, middle, replace 'access' with 'excess'

Fig. 5 B and C. Explain in the legend that 'FM' stands for 'Factor Mix'.

p.15, l. 7. Replace "slowly" with 'slower'

How do authors explain that THB does not stimulate association of 30S IC with the 50S subunit (Suppl Fig 1B), but does stimulate association of the vacant subunits (Fig. 5D)?

Fig. 6B. The representation of RNA in the panel B is confusing. Is it the cryo-EM density that is shown or the built model? Why some RNA bases are 'thick' (shown as blobs) and others are thin (shown as sticks)?

p.19 and throughout: The use of "AGA" as an abbreviation for aminoglycosides is fairly unconventional and might confuse the reader. I would recommend using 'aminoglycoside' throughout.

Materials and methods section needs to be amended with the proper references and the reagent

sources. Thus, for example, the sentence "Escherichia coli MRE600 cells were used to prepare tight-coupled 70S ribosomes following standard laboratory protocol" makes it difficult to reproduce the described experiments.

Reviewer #2 (Remarks to the Author):

In this manuscript, Parajuli et al. lay out a proposed mechanism of action for the antibiotic thermorubin (THB), which binds to the B2a subunit bridge of the bacterial ribosome. Contrary to previous studies suggesting that THB inhibits initiation, the authors show that it in fact impedes elongation, termination, and ribosome recycling. They illustrate this through a series of rapid kinetic and structural experiments. The experiments are well-designed, thoroughly performed, and mostly adequately controlled. Although the data seem to have been thoughtfully analyzed, the descriptions of how most of the data were analyzed were not detailed enough such that the quality of the analyses could be determined. The mechanistic finding is novel and resolves previous results and theories on the mechanism of inhibition by THB that were quite confusing. Thus, I recommend this manuscript for publication in Nature Communications subject to the following revisions:

1. The manuscript is written in a way that only targets a small segment of highly-specialized readers working in the field of translation. It assumes prior knowledge of the general mechanisms of ribosomal initiation, elongation, termination, and ribosome recycling pathways, and thus the kinetic inferences made using this assumption of prior knowledge is likely to induce a high level of confusion in a broader audience. The authors should consider revising the manuscript in such a way as to make it more accessible to a broader audience. For example, the basic mechanisms involved in ternary complex binding, peptidyl transfer, subunit rotation, translocation, and the binding of the factors involved in all stages of translation (especially given that the function of these factors contributes to the authors' mechanistic hypothesis), etc., should be defined. Perhaps the authors can prepare an SI figure depicting and defining the mechanistic steps of translation initiation, elongation, termination, and ribosome recycling that are relevant to their kinetic and biochemical assays (something similar to their Figure 7, but focused on the mechanistic steps of translation initiation, elongation, termination, and ribosome recycling that are relevant to their kinetic and biochemical assays, rather than on the steps of translation that THB inhibits).

2. Significant clarifications are needed concerning the rapid kinetics experiments, analyses, and interpretations presented in the manuscript. Specifically, the following questions need to be addressed:

- a. The rapid kinetics experiments and their analyses and interpretations presented in the current work are not described in enough detail to allow me (or any reader, actually) to assess how rigorously they were performed, analyzed, or interpreted. Thus, it is difficult to determine how strongly the results support the conclusions that are drawn. While I appreciate that most (all?) of the rapid kinetics experiments, analyses, and interpretations performed in the current study have been described in previous publications (e.g., references 17, 19-23, 26), the current manuscript lacks entirely too much detail and is nearly impossible to assess as a standalone work. This will be particularly true for the readership of a general science journal such as Nature Communications. Consider as just one example, how the authors extract the mean time for translocation from the results of rapid kinetics experiments measuring the mean times of di- and tripeptide formations. While it may be reasonable to expect that an expert on rapid kinetics studies of translation will understand why the mean time of translocation can be roughly estimated by subtracting the mean times of two dipeptide formations from the total time for tripeptide formation, this is not further explained or adequately defended in a way that would make sense to a broader audience. This example highlights an issue that pervades all of the rates presented in the manuscript, and a clear explanation and reasoning for how each rate is derived need to be presented. Additionally, the derivations, equations, and calculations that were used

to obtain each rate need to be put into the Supplementary Information for reference; merely reporting that the signals were fit to exponential functions is not a description of the data analyses that can be assessed.

b. The rates of THB association and dissociation to ribosomal complexes are implicated, assumed, and/or estimated during the analysis and interpretation of several of the rapid kinetics experiments. Given the importance these rates take on in the analysis and interpretation of the data, I wonder whether they have been previously established or published and, if not, whether there is a straightforward way for the authors to directly measure them?

c. The rate of THB dissociation is estimated to be 0.02s^{-1} based on the rate of peptide release in the presence of THB at saturating RF2 concentration. This makes several assumptions that are not adequately defended in the manuscript. For example, tying in with points a and b above, there is a blanket assumption that THB is bound tightly throughout the entire elongation cycle and does not dissociate regardless of subunit rotation. Additionally, it is stated that RF2 can only bind weakly in the presence of THB, and only upon spontaneous dissociation can it extend and facilitate peptide hydrolysis. However, no direct evidence of either of these things is presented, and caveats for these hypotheses are not given.

3. There is a discrepancy between statements made regarding the affinity of THB for ribosomal subunits and ribosomal complexes in the context of initiation versus termination. For the initiation experiments, it is stated that THB does not bind the individual subunits tightly and, in fact, has no effect on subunit association. However, in the recycling experiments, it is stated that THB facilitates reassociation of split subunits; this implies that the THB is not only tightly bound to an individual subunit, but does in fact affect subunit association. Is this simply an effect of the presence of different factors during initiation and ribosome recycling? The authors have adequately shown that THB inhibits dissociation of ribosomal subunits during recycling, but the same cannot be said for the assumption that it also induces reassociation of split subunits. In my view, the experiments shown in Figures 5A-5C represent complex equilibria that are likely not as easily interpreted as the authors suggest. The red curves in Figures 5A and 5C comprise the data recorded when THB is added to pre-TCs. The slow decrease in the signal is interpreted as a slow splitting of the subunits. In reality, it is likely that this slowly decreasing signal represents a dynamic equilibrium between the low probability of FM-driven splitting of THB-free ribosomes and the high probability of subunit reassociation into ribosomes that are then bound by THB and stabilized. Similarly, the red curve in Figure 5B and the green curve in Figure 5C comprise the data recorded when THB is added to the pre-TCs in competition with the FM. The fast decrease in half the amplitude of the signal is interpreted as a fast FM-driven splitting of THB-free ribosomes, and the subsequent slow increase of the signal is interpreted as a THB-driven reassociation of the split subunits. In reality, it is likely that this slowly increasing signal represents the same dynamic equilibrium described above with the caveat that the pre-TCs were not pre-saturated with THB. This effect can perhaps best be seen in Figure 5C, where the red and green curves seem to be converging to the same equilibrium position. Perhaps if the experiment had been allowed to run another few hundred seconds, they would have fully converged.

4. While the authors have shown that THB inhibits the elongation, termination, and ribosome recycling stages of translation, the structure they have solved is of an initiation complex. This adequately shows that THB does not disrupt the binding of fMet-tRNA^{fMet} and that it induces the rearrangement of certain ribosomal residues, but it is not relevant for discussing the role of THB in inhibiting the elongation, termination, and ribosome recycling stages of translation. Given that the authors do not have structures of THB bound to ribosomal complexes engaged in elongation, termination, and ribosome recycling, it would be helpful and potentially very insightful for the authors to perform comparative analyses in which they compare their structure with existing structures of ribosomal complexes engaged in the elongation, termination, and ribosome recycling stages of translation. For example, would the superimposition of a ternary complex, RF, or RRF from an existing structure of a ribosomal complex exhibit any steric clashes or other obvious incompatibilities with the THB-bound

initiation complex structure presented in the current work? Additionally, the THB-bound initiation complex structure presented in the current work should be compared to structures of ribosomal complexes bound to antibiotics such as aminoglycosides, viomycin, tetracycline, etc. and variations in the disruption of important ribosomal residues should be discussed.

5. Careful proofreading for grammar and typos is needed, as there are many errors throughout the text. Additionally, figure callouts throughout the text need to be reviewed and assessed for correctness, as there are several issues. Following are several examples, but there are many more:

a. Page 7 states that Figure 2C reports the results of BOP-Met-tRNA^{fMet} in the presence of various combinations of IFs, however, the figure simply shows with/without IFs.

b. Page 8 states Figure 3A as reporting on (tau)GTP and (tau)Dip in the absence of THB and Figure 3B as reporting on (tau)GTP and (tau)Dip in the presence of THB. However, Figure 3A reports on GTP hydrolysis in the presence and absence of THB, while Figure 3B reports on dipeptide formation in the presence and absence of THB.

c. Page 9 states Figure S4A shows the case where THB was added with ternary complex to THB-free ribosomes, however, the figure legend states that THB was pre-incubated with 70S ribosomes at the end of the description.

d. Page 15 and the Methods and Materials section state that both IF3 and IF1 were used for experiments with data shown in Figures 5C and D. However, the legend for Figure 5D says that the data shown for THB in FM with IFs (grey line) only includes IF3.

Reviewer #3 (Remarks to the Author):

One approach to increasing the number of compounds entering the antibiotic development pipeline is to re-examine old or neglected antibiotics that are considered unsuitable for therapeutic use, and modify them to make new effective drugs. In this work, Parajuli et al. use fast kinetics and cryo-EM to characterize the mechanism of action of Thermorubin (THB), a broad-spectrum antibiotic generally considered unsuitable for human or veterinary treatment due to its low solubility. They reveal that, contrary to what had previously been suggested, THB does not inhibit the formation of the translation initiation complex, but rather interferes with multiple steps of the translation cycle (aminoacyl-tRNA accommodation, peptide release, ribosome recycling and subunit dissociation). Finally, they propose that a flipped-out conformation of 23S rRNA base C1914 induced by THB binding is likely responsible for these effects. Detailed characterization of this type is important because it will help guide the development of THB variants with greater solubility that retain some or all of the inhibitory mechanisms of the parent compound.

Overall, this is a timely and well-crafted study that sheds light on the mechanism of action of a neglected, but potentially useful class of antibiotics. The kinetic characterization is detailed and thoroughly executed, and the conclusions are well supported by the data.

My only major comment concerns the cryo-EM section, specifically the claimed resolution of the structure, which appears to be overestimated. Indeed, the overall resolution of the reconstruction is estimated to be 1.96 Å, which, if correct, would be higher than all but one published ribosome structure (Watson et al. eLife 2020;9:e60482). The data were processed using CryoSparc, which generally gives a somewhat higher resolution estimate than Relion, but I am inclined to think that this is not the only reason for the claimed high resolution value. In particular, there are a few things that I find surprising: (1) the local resolution is very high and unusually homogeneous over the entire structure, including the most mobile parts of the ribosome, such as the small subunit or the L1 stalk

(compare Supp. Fig. 9 with Fig. 1a of Watson et al. eLife 2020); (2) although it is difficult to judge from the few density images provided by the authors, the map resolution looks more like 2.3-2.5 Å than 1.96 Å in my opinion; for example, distinct density is clearly visible for the -O-Me and -C=O groups of THB, but not for the -OH groups (Fig. 6); moreover, one would expect to start seeing holes in the density for the THB tetracycle rings at the claimed resolution.

I would therefore recommend that the authors carefully check or reprocess their cryo-EM data using Cryosparc and/or Relion to ensure that the map quality is in agreement with the reported resolution. In particular, they should look out for duplicate particles in their dataset, which would inevitably lead to an overestimated resolution. Finally, the authors should show additional densities and discuss features of the reconstruction that are compatible with the resolution of their map. This might include isolated ligand densities, solvent molecules, tRNA modifications, etc.

Besides this, I have the following minor comments:

- In Fig. 3E the label at the top should read "and" instead of "aand".
- In Fig. 6D, the density for THB is hard to see. If solvent molecules coordinating the drug can be seen it is important to show them in this panel and, if possible, to avoid carving the density around the model.
- On p. 18, it is stated that "The repositioned C1914 sterically interferes with the accommodating aa-tRNA in the A-site, explaining both impaired binding and increased drop-off in the presence of THB." A structural comparison between the structure presented in this work and those of intermediates observed during A-tRNA delivery by EF-Tu and its subsequent accommodation (see studies from the Korostelev lab) would make a useful addition to Fig. 6 by giving the reader an idea of the extent of the clashes induced by the drug during aminoacyl-tRNA accommodation.
- Similarly, on p. 20 - "The THB effect on termination originates most probably from THB-induced flipping of the monitoring bases A1492 and A1493 out of h44 (Fig. 6) which would impede RF binding and stop codon recognition since the flipped A1493 would clash with the domain-II of RF". A panel showing potential clashes with RFs would be a good addition to Fig. 6.
- Are there mutations that confer THB resistance? If so, where are they located and how does this fit with the proposed mechanism? A few lines to this effect could be added to the discussion section.
- In Supp. Fig. 1, could the authors comment on the different kinetics of subunit association in the absence of IFs with or without THB?
- Heterogeneous 3D classification in Cryosparc yielded only two classes (Supp. Fig. 9): one consisting of 70S particles (96.1% of "good" particles) and the other predominantly of 50S particles (3.9%). Given the nature of the complex, one might also expect empty 70S ribosomes. Could the authors comment on the occupancy of fMet-tRNA^{iMet} in their complex and whether densities for the ribosome and tRNA appear to be stoichiometric? It is also important that they use the same contour levels to display densities for the tRNA, ribosome and mRNA in Fig. 6A and 6B, and that they use unmasked maps in which all density is shown for both of these panels (not just density around the model). This should be clearly specified in the figure legend.
- A structural comparison with the earlier X-ray structure from the Steitz lab would be helpful, possibly as a supplementary figure. Did the gain in resolution reveal additional contacts or notable differences?

Point-by-point response to the reviewer's comments

Reviewer's comments – in black, our response – in red

Reviewer #1 (Remarks to the Author):

The paper of Parajuli et al. explores the mechanism of action of the ribosome-targeting antibiotic thermorubin, a long-known drug but with an unclear mechanism of action. BY running an array of kinetic assays and obtaining a high-resolution structure of the antibiotic bound to the ribosomal functional complex, the authors demonstrate that the drug interferes with accommodation of aminoacyl-tRNA in the ribosomal A site and with the action of class I release factors and ribosome recycling factor. As the result, in contrast to the previous assertion that thermorubin inhibits initiation of translation, the authors conclude that the drug in fact interferes with translation elongation, termination and recycling.

- **Response:** We thank the reviewer for the concise and yet to-the-point summary of our work.

Critique

It is an interesting and well performed study. The experiments are well designed and well described, the results are convincing, and the conclusions are reasonable. The paper is well written and is an easy read.

My only main problem is that discussing the structure of the ribosome-antibiotic complex, the authors do not analyze possible clash of thermorubin with the A-site tRNA, RF or RRF. On p.16. (ll. 7-8) they allude that bases C1914 and U1915 'occupy the region of the vacant A site'. Do imply that there a direct clash with "aminoacyl-tRNA, class-I RFs, and RRF"? If so, how bad is the clash? If not, how do they explain their data? Any of these scenarios need to be properly illustrated using the available relevant structures and disussed in specific terms rather than using vague general statements.

- **Response:** We thank the reviewer for this insightful comment. As the reviewer suggested, we have analyzed the possibility of molecular clash between the displaced bases by THB binding and the A-site substrates. For that, we re-analyzed our structure (PDB 8AYE) in comparison with available structures of elongating (aa-tRNA bound) ribosomes (PDB 7K00), RF1 bound termination complex (PDB 6DNC), EF-G bound pre-translocation ribosome (PDB 7SSL) and RRF bound post-termination ribosome (PDB 4V54) and added the analyses in new Fig. 6D – F, Supplementary Figure S15.

As shown in the modified Figure 6, the displaced base C1914 shows moderate and severe clash with aa-tRNA (Fig. 6D) and domain II residues of RF1, respectively (Fig. 6E). In fact, the clash with RF1 is significant involving multiple residues. Overlay of RRF structure on the other hand did not show any possibility of molecular clash (Fig. 6F), which is why we have adjusted our explanations regarding the mechanism of occlusion of RRF binding (page 14 main text, highlighted). We propose alternatively that the stabilization of the ribosome in classical/non-rotated state can be the possible reason for inhibition of RRF binding. More importantly, the 'gluing' effect of THB is the primary reason for inhibition of ribosome recycling. This has been detailed in the results (page 14 main text, highlighted) and

discussion (page 17 main text, highlighted) sections. As suggested by the reviewer illustrations have been added (Figs. 6D – F, Supplementary Fig. S16) and abstract is also updated (see highlighted text).

The other problems are minor:

p.8, top. besides three steps of the elongation cycle mentioned by the authors, it also includes peptide bond formation.

- **Response:** Thanks to the referee for mindful checking. It has now been included as the following. ‘In parallel, we followed dipeptide (fMet-Phe) formation, which includes, in addition to the delivery, aa-tRNA accommodation and subsequent peptidyl-transfer reaction’ (page 7, main text highlighted).

Fig. 3E. correct typo in the text line above the graph

- **Response:** The typo in Fig. 3E has been corrected.

p.12, bottom. It was unclear to me where the conclusion that THB needs to dissociate from the ribosome for peptidyl-tRNA hydrolysis to take place came from. What data show that RF cannot promote peptidyl-tRNA hydrolysis on the THS-bound ribosome, albeit slowly?

- **Response:** We thank the reviewer for raising this important question. Following this question we re-analyzed our peptide release kinetics data, which actually can be explained with two models. While it is compatible with the model that peptide release by RF1/RF2 can happen in a highly defective manner while THB is bound to the ribosome, our data can also be explained with an alternative model, which requires that THB should dissociate for RF1/RF2 to bind and release peptide actively.

To clarify this, we use the structural analysis (suggested by reviewer 1 and 2) to decide the most plausible model between the two. Overlay of our THB-bound ribosome structure with RF1-bound termination complex (PDB 6DNC) shows severe clashes between multiple residues of the domain II of RF1 and the displaced base C1914 from 23S rRNA (Fig. 6E). The clashes are much more severe than what we observe for aa-tRNA (Fig. 6D) (also pasted below). Thus, most likely RF1 binding and accommodation would be super challenging by THB-induced structural changes in the A site. This is why, although we present both the

models in the discussion, we support more the THB dissociation model.

Fig 6D : Overlay with aa-tRNA bound structure and 6E with RF1 bound structure of the ribosome

In addition, we have now measured the association and dissociation kinetics of THB using stopped-flow fluorescence. We find 0.003 s^{-1} as the maximal value for the dissociation rate constant of THB from 70S or pre-termination complex. It means that spontaneous THB dissociation takes on-average no less than 5 min (Supplementary Fig. S6). Compared to that

only 45 s is required for peptide release by RF2 in the presence of THB (Fig. 4D, inset). This comparison implies that pre-bound RF can destabilize THB and hasten THB dissociation for successful peptide release. This analysis is now added to Supplementary text and discussed in the modified result (page 11 main text, highlighted) and discussion (page 17 main text, highlighted) (see also our reply to Reviewer 2).

p.14, middle, replace ‘access’ with ‘excess’

- **Response:** Thanks to the reviewer for pointing out this typo. It is now changed to ‘excess’.

Fig. 5 B and C. Explain in the legend that ‘FM’ stands for ‘Factor Mix’.

- **Response:** The figure legends are updated with full form – Factor Mix.

p.15, l. 7. Replace “slowly” with ‘slower’

- **Response:** The word ‘slowly’ is changed to slower.

How do authors explain that THB does not stimulate association of 30S IC with the 50S subunit (Suppl Fig 1B), but does stimulate association of the vacant subunits (Fig. 5D)?

- **Response:** It is known that THB binds very weakly to the individual subunits. Our experiment shows that THB binding to 70S takes very long time ($k_a = 0.03/s$). Presumably THB binding to the subunits will take longer. Now, the association of 30S pre-IC (without IF3) with 50S is very rapid (~1 s) (Fig. S1B) and thus insufficient for THB binding. Hence, the fact that we do not see any effect of THB in proper subunit association, particularly without IF3 is not surprising. When it comes to association of the free subunits split by lowering Mg^{+2} concentrations (Fig. 5D), it is a very slow 70S forming reaction (takes about 5 min) due to constant forward (association) and backward (dissociation) reactions. Only when 70S associates very slowly from the subunits THB binding occurs and that moves the equilibrium towards formation of 70S. That is why it apparently seems that THB stimulates 70S formation from the free subunits. In reality, THB blocks dissociation of the chance-associated 70S and thus pushes the equilibrium to towards re-formed 70S. Thus, the results in Supplementary Fig. S1B are not contradictory to that in Fig. 5D.

Fig. 6B. The representation of RNA in the panel B is confusing. Is it the cryo-EM density that is shown or the built model? Why some RNA bases are ‘thick’ (shown as blobs) and others are thin (shown as sticks)?

- **Response:** Now we have updated the figure 6. The densities in 6A show represent the model fitted to the map (together). Issues with earlier panel 6B is now resolved. The earlier 6B with zoom-in binding pocket of THB has been moved to a Supplementary Figure S13. There we show both model + map (A) and map alone (B). The same contour level was used throughout the figure including tRNA, mRNA, rRNA and THB. The equal and high resolution of the components almost makes the map as good as the one with the model.

p.19 and throughout: The use of “AGA” as an abbreviation for aminoglycosides is fairly unconventional and might confuse the reader. I would recommend using ‘aminoglycoside’ throughout.

- **Response:** We have updated the text as per the reviewer's suggestions and replaced AGAs with aminoglycosides.

Materials and methods section needs to be amended with the proper references and the reagent sources. Thus, for example, the sentence "Escherichia coli MRE600 cells were used to prepare tight-coupled 70S ribosomes following standard laboratory protocol" makes it difficult to reproduce the described experiments.

- **Response:** We have updated the materials and methods section with appropriate reference (Johansson et al., 2008).

Reviewer #2 (Remarks to the Author):

In this manuscript, Parajuli et al. lay out a proposed mechanism of action for the antibiotic thermorubin (THB), which binds to the B2a subunit bridge of the bacterial ribosome. Contrary to previous studies suggesting that THB inhibits initiation, the authors show that it in fact impedes elongation, termination, and ribosome recycling. They illustrate this through a series of rapid kinetic and structural experiments. The experiments are well-designed, thoroughly performed, and mostly adequately controlled. Although the data seem to have been thoughtfully analyzed, the descriptions of how most of the data were analyzed were not detailed enough such that the quality of the analyses could be determined. The mechanistic finding is novel and resolves previous results and theories on the mechanism of inhibition by THB that were quite confusing. Thus, I recommend this manuscript for publication in Nature Communications subject to the following revisions:

- **Response:** We thank the reviewer for taking time to meticulously evaluate the importance of our results presented in this manuscript. We are very happy to see the meticulous analysis of our biochemical results by the reviewer. We have addressed all concerns of the reviewer, which led to much higher quality of this manuscript.

1. The manuscript is written in a way that only targets a small segment of highly-specialized readers working in the field of translation. It assumes prior knowledge of the general mechanisms of ribosomal initiation, elongation, termination, and ribosome recycling pathways, and thus the kinetic inferences made using this assumption of prior knowledge is likely to induce a high level of confusion in a broader audience. The authors should consider revising the manuscript in such a way as to make it more accessible to a broader audience. For example, the basic mechanisms involved in ternary complex binding, peptidyl transfer, subunit rotation, translocation, and the binding of the factors involved in all stages of translation (especially given that the function of these factors contributes to the authors' mechanistic hypothesis), etc., should be defined. Perhaps the authors can prepare an SI figure depicting and defining the mechanistic steps of translation initiation, elongation, termination, and ribosome recycling that are relevant to their kinetic and biochemical assays (something similar to their Figure 7, but focused on the mechanistic steps of translation initiation, elongation, termination, and ribosome recycling that are relevant to their kinetic and biochemical assays, rather than on the steps of translation that THB inhibits).

- **Response:** We think that this is a valuable suggestion and adding the schemes suggested by the reviewer will increase readership of the paper. We have thus added the required schemes of ribosome initiation and elongation cycle in SI text together with schematics. We have also added detailed derivations for the powerful mean time analysis method we established in this work. Although mean time analysis has been used by us in earlier works, now the compiled SI text can be used for the method for mean time analysis in all major steps of translation.

2. Significant clarifications are needed concerning the rapid kinetics experiments, analyses, and interpretations presented in the manuscript. Specifically, the following questions need to be addressed:

- **Response:** We understand the concern of the reviewer and have acted accordingly. Please see our responses below.

a. The rapid kinetics experiments and their analyses and interpretations presented in the current work are not described in enough detail to allow me (or any reader, actually) to assess how rigorously they were performed, analyzed, or interpreted. Thus, it is difficult to determine how strongly the results support the conclusions that are drawn. While I appreciate that most (all?) of the rapid kinetics experiments, analyses, and interpretations performed in the current study have been described in previous publications (e.g., references 17, 19-23, 26), the current manuscript lacks entirely too much detail and is nearly impossible to assess as a standalone work. This will be particularly true for the readership of a general science journal such as Nature Communications.

Consider as just one example, how the authors extract the mean time for translocation from the results of rapid kinetics experiments measuring the mean times of di- and tripeptide formations. While it may be reasonable to expect that an expert on rapid kinetics studies of translation will understand why the mean time of translocation can be roughly estimated by subtracting the mean times of two dipeptide formations from the total time for tripeptide formation, this is not further explained or adequately defended in a way that would make sense to a broader audience. This example highlights an issue that pervades all of the rates presented in the manuscript, and a clear explanation and reasoning for how each rate is derived need to be presented. Additionally, the derivations, equations, and calculations that were used to obtain each rate need to be put into the Supplementary Information for reference; merely reporting that the signals were fit to exponential functions is not a description of the data analyses that can be assessed.

- **Response:** We have addressed the reviewer's concern by adding a Supplementary section that describes in details the basics of "mean time" analysis from the first principles of enzyme kinetics, i.e. starting from differential equation describing the dynamics of concentration of different ribosomal complexes for all kinetic experiments performed in this paper. The SI contains individual sections for initiation, elongation and termination, which are supported by schematic description of the pathways with sub-steps. In addition, we also explain in SI how the mean time analysis is used to derive relevant rate constants in kinetic experiments performed in this work. All the derivations, equations, and calculations that were used to obtain each rate have been added in the supplementary. Thus, we believe that this paper can now be read as a standalone work. In fact, this near-complete kinetic model analysis (SI text)

can be largely used and cited in future papers, which will use mean time analysis for mechanistic understanding of complex molecular processes. We choose not to add these detailed data-fitting equations in the main text not to overwhelm the general readers with technicalities of kinetic experiments.

b. The rates of THB association and dissociation to ribosomal complexes are implicated, assumed, and/or estimated during the analysis and interpretation of several of the rapid kinetics experiments. Given the importance these rates take on in the analysis and interpretation of the data, I wonder whether they have been previously established or published and, if not, whether there is a straightforward way for the authors to directly measure them?

- **Response:** The affinities of THB to naked subunits and to 70S ribosome have been estimated (Lin et. al., 1982). However, to our knowledge, the THB association/dissociation rates to and from the ribosome have not been measured before. Following the reviewer's suggestion we measured the second order association rate constant for THB binding to 70S ribosomes in a stop-flow experiment using the fact that THB fluorescence changes upon its binding to the ribosome (Lin et. al., 1982). The measured association rate constant, $k_a=0.03 \mu\text{M}^{-1}\text{s}^{-1}$, confirms our previous assertion that at $10 \mu\text{M}$ THB, the highest THB concentration used in most of our experiments here, THB binds ribosomes much slower than ternary complexes, EF-G and other translation factors. We have also estimated the upper bound for THB dissociation rate constant as 0.0027 s^{-1} in experiments where we chased ribosome-bound THB from the 70 ribosome and 70S pre-termination complexes by high concentration of non-fluorescent antibiotic arbekacin that is known to bind tightly to the ribosome, to the binding site overlapping with that of THB so that Arb and THB binding is mutually exclusive. The chase in this case was not perfect since in equilibrium a considerable THB fraction was still bound to the ribosome indicating that THB dissociation rate constant should be much smaller than 0.0027 s^{-1} . These stop-flow experiments (together with their description) are added now as Supplementary Fig. S5 and S6. We also discuss the kinetic equation for complex formation and chase experiments in the kinetic section of SI text. We have also added these results in the main text (page 8, main text, highlighted).

c. The rate of THB dissociation is estimated to be 0.02s^{-1} based on the rate of peptide release in the presence of THB at saturating RF2 concentration. This makes several assumptions that are not adequately defended in the manuscript. For example, tying in with points a and b above, there is a blanket assumption that THB is bound tightly throughout the entire elongation cycle and does not dissociate regardless of subunit rotation. Additionally, it is stated that RF2 can only bind weakly in the presence of THB, and only upon spontaneous dissociation can it extend and facilitate peptide hydrolysis. However, no direct evidence of either of these things is presented, and caveats for these hypotheses are not given.

- **Response:** We do not agree with the reviewer that our assertion that THB, once bound, stays on the ribosome during the whole elongation cycle is “a blanket assumption”. This assertion comes from our analysis of di- and tri-peptide experiments in which we observed that each additional peptide bond formation is equally ($\sim 0.5 \text{ sec}$) slow in the presence of THB. We observe, on the other hand, that when THB is added together with ternary complex to the

ribosomes, the peptide bond formation occurs fast, with the rate observed in the absence of THB. Then, if THB were dissociated during or before the first translocation in di- or tri-peptide formation experiment one would expect its re-binding outcompeted by T_3 and a fast formation of the second and third peptide bond.

We now discuss in more detail (see modified MS text and SI text) different scenarios of peptide release including the scenario in which the release is occurring in the presence of THB on the ribosome. However, we favor the alternative scenario based on structural analysis, which shows severe clashes of the displaced C1914 with the domain II residues of RF1. Please see more details in our reply with illustration to reviewer 1 to his/her remark “p.12, bottom”.

3. There is a discrepancy between statements made regarding the affinity of THB for ribosomal subunits and ribosomal complexes in the context of initiation versus termination. For the initiation experiments, it is stated that THB does not bind the individual subunits tightly and, in fact, has no effect on subunit association. However, in the recycling experiments, it is stated that THB facilitates reassociation of split subunits; this implies that the THB is not only tightly bound to an individual subunit but does in fact affect subunit association. Is this simply an effect of the presence of different factors during initiation and ribosome recycling? The authors have adequately shown that THB inhibits dissociation of ribosomal subunits during recycling, but the same cannot be said for the assumption that it also induces reassociation of split subunits. In my view, the experiments shown in Figures 5A-5C represent complex equilibria that are likely not as easily interpreted as the authors suggest. The red curves in Figures 5A and 5C comprise the data recorded when THB is added to pre-TCs. The slow decrease in the signal is interpreted as a slow splitting of the subunits. In reality, it is likely that this slowly decreasing signal represents a dynamic equilibrium between the low probability of FM-driven splitting of THB-free ribosomes and the high probability of subunit reassociation into ribosomes that are then bound by THB and stabilized. Similarly, the red curve in Figure 5B and the green curve in Figure 5C comprise the data recorded when THB is added to the pre-TCs in competition with the FM. The fast decrease in half the amplitude of the signal is interpreted as a fast FM-driven splitting of THB-free ribosomes, and the subsequent slow increase of the signal is interpreted as a THB-driven reassociation of the split subunits. In reality, it is likely that this slowly increasing signal represents the same dynamic equilibrium described above with the caveat that the pre-TCs were not pre-saturated with THB. This effect can perhaps best be seen in Figure 5C, where the red and green curves seem to be converging to the same equilibrium position. Perhaps if the experiment had been allowed to run another few hundred seconds, they would have fully converged.

- **Response:** The explanation to this apparent discrepancy is that when 70S ribosome is formed by subunit association, a strong THB binding site on 70S is created. It then follows from purely thermodynamic considerations that the subsequent THB binding to the 70S stabilizes it, i.e. decreases the dissociation rate of subunits. This can very well occur without THB affecting the association rate constant of the ribosomal subunits as indeed observed in our “initiation experiments”. This is also quite in line with previous measurements (Lin et. al., 1982) showing very weak THB binding to individual subunits (i.e. if THB does not bind to subunits no THB effect on their association is expected).

In recycling experiments, the initially THB-free ribosomes dissociated into subunits by RRF/EF-G, IF1/IF3 or by lowering Mg^{++} will eventually form an empty 70S ribosome when dissociation effect by RRF/EF-G or IF1/IF3 will be inefficient. Such 70S ribosomes will eventually bind THB and become protected from dissociation against all mentioned factors (RRF/EF-G, IF1/IF3, Mg^{++}). Thus, we removed the statements claiming that ‘THB facilitates / induces reassociation of split subunits’. Instead we have carefully revised the text both for Figs. 5B, C and D. Please see the highlighted text in pages 12 and 13. We however agree with the reviewer that the approach to attaining equilibrium can be complex, especially in RRF/EF-G case where GTP hydrolysis complicates the thermodynamic analysis. But in other cases, the genuine equilibrium will eventually be reached, and, as the reviewer pointed out, perhaps, would have been seen if the measurement was continued for a longer time in our 5C experiments.

4. While the authors have shown that THB inhibits the elongation, termination, and ribosome recycling stages of translation, the structure they have solved is of an initiation complex. This adequately shows that THB does not disrupt the binding of fMet-tRNA^{fMet} and that it induces the rearrangement of certain ribosomal residues, but it is not relevant for discussing the role of THB in inhibiting the elongation, termination, and ribosome recycling stages of translation. Given that the authors do not have structures of THB bound to ribosomal complexes engaged in elongation, termination, and ribosome recycling, it would be helpful and potentially very insightful for the authors to perform comparative analyses in which they compare their structure with existing structures of ribosomal complexes engaged in the elongation, termination, and ribosome recycling stages of translation. For example, would the superimposition of a ternary complex, RF, or RRF from an existing structure of a ribosomal complex exhibit any steric clashes or other obvious incompatibilities with the THB-bound initiation complex structure presented in the current work? Additionally, the THB-bound initiation complex structure presented in the current work should be compared to structures of ribosomal complexes bound to antibiotics such as aminoglycosides, viomycin, tetracycline, etc. and variations in the disruption of important ribosomal residues should be discussed.

- **Response:** We are thankful to the reviewer for the comment. We have addressed it properly and added new figures. Please see our response to the reviewer 1 (p.12, bottom...)

5. Careful proofreading for grammar and typos is needed, as there are many errors throughout the text. Additionally, figure callouts throughout the text need to be reviewed and assessed for correctness, as there are several issues. Following are several examples, but there are many more:

- **Response:** We have carefully reviewed the figures references in the text. In agreement with the reviewer, there were multiple errors, which we have fixed. We also thank the reviewer to point out the typos and punctuation errors. The manuscript is now language-edited by neutral readers, who are native speakers.

a. Page 7 states that Figure 2C reports the results of BOP-Met-tRNA^{fMet} in the presence of various combinations of IFs, however, the figure simply shows with/without IFs.

- **Response:** We have updated the text and figure legends for Fig. 2C (with IFs) and 2D (without IFs).

b. Page 8 states Figure 3A as reporting on (tau) GTP and (tau)Dip in the absence of THB and Figure 3B as reporting on (tau)GTP and (tau)Dip in the presence of THB. However, Figure 3A reports on GTP hydrolysis in the presence and absence of THB, while Figure 3B reports on dipeptide formation in the presence and absence of THB.

- **Response:** The corresponding text for figures 3A and 3B is corrected (page 7, main text highlighted).

c. Page 9 states Figure S4A shows the case where THB was added with ternary complex to THB-free ribosomes, however, the figure legend states that THB was pre-incubated with 70S ribosomes at the end of the description.

d. Page 15 and the Methods and Materials section state that both IF3 and IF1 were used for experiments with data shown in Figures 5C and D. However, the legend for Figure 5D says that the data shown for THB in FM with IFs (grey line) only includes IF3.

- **Response:** We thank the reviewer for these careful observations. The text in the corresponding figure legends (both as commented in c. and d.) has been updated with the correct statements.

Reviewer #3 (Remarks to the Author):

One approach to increasing the number of compounds entering the antibiotic development pipeline is to re-examine old or neglected antibiotics that are considered unsuitable for therapeutic use and modify them to make new effective drugs. In this work, Parajuli et al. use fast kinetics and cryo-EM to characterize the mechanism of action of Thermorubin (THB), a broad-spectrum antibiotic generally considered unsuitable for human or veterinary treatment due to its low solubility. They reveal that, contrary to what had previously been suggested, THB does not inhibit the formation of the translation initiation complex, but rather interferes with multiple steps of the translation cycle (aminoacyl-tRNA accommodation, peptide release, ribosome recycling and subunit dissociation). Finally, they propose that a flipped-out conformation of 23S rRNA base C1914 induced by THB binding is likely responsible for these effects. Detailed characterization of this type is important because it will help guide the development of THB variants with greater solubility that retain some or all of the inhibitory mechanisms of the parent compound.

Overall, this is a timely and well-crafted study that sheds light on the mechanism of action of a neglected, but potentially useful class of antibiotics. The kinetic characterization is detailed and thoroughly executed, and the conclusions are well supported by the data.

- **Response:** We thank the reviewer for lifting up the importance of our current study from the perspective of antibiotic resistance problem and global trend of repurposing old antibiotics.

My only major comment concerns the cryo-EM section, specifically the claimed resolution of the structure, which appears to be overestimated. Indeed, the overall resolution of the reconstruction is estimated to be 1.96 Å, which, if correct, would be higher than all but one published ribosome structure (Watson et al. eLife 2020;9:e60482). The data were processed using CryoSparc, which generally gives a somewhat higher resolution estimate than Relion,

but I am inclined to think that this is not the only reason for the claimed high resolution value. In particular, there are a few things that I find surprising: (1) the local resolution is very high and unusually homogeneous over the entire structure, including the most mobile parts of the ribosome, such as the small subunit or the L1 stalk (compare Supp. Fig. 9 with Fig. 1a of Watson et al. eLife 2020); (2) although it is difficult to judge from the few density images provided by the authors, the map resolution looks more like 2.3-2.5 Å than 1.96 Å in my opinion; for example, distinct density is clearly visible for the -O-Me and -C=O groups of THB, but not for the -OH groups (Fig. 6); moreover, one would expect to start seeing holes in the density for the THB tetracycline rings at the claimed resolution.

I would therefore recommend that the authors carefully check or reprocess their cryo-EM data using Cryosparc and/or Relion to ensure that the map quality is in agreement with the reported resolution. In particular, they should look out for duplicate particles in their dataset, which would inevitably lead to an overestimated resolution. Finally, the authors should show additional densities and discuss features of the reconstruction that are compatible with the resolution of their map. This might include isolated ligand densities, solvent molecules, tRNA modifications, etc.

- **Response:** We thank reviewer for this detailed comment. We divide our responses under multiple subheadings.

Duplication of particles: We now have re-evaluated the processing pipeline of the cryo-EM data. We have particularly re-run cryoSPARC based checking of the duplicate particles in our reconstruction and found that there was no duplication of particles. Further, we ran a Relion pipeline till 3D classification, which also had more than more than 2 M particles. Thus, we are certain that there was no particle duplication and that the high number of particles led to such a high resolution.

Local resolution: We thank the expert reviewer for his comment about local resolution of the mobile parts. Following the comment, we have analyzed our map with another round of ResMap analysis. Why it still shows large section of homogeneous high density, we do see much better distribution of the resolution over the structure. Moreover, we do see lower resolution in the mobile parts such as the L1 stalk and parts of the small subunit. Thus the cryo-EM pipeline image has been updated (Supplementary Fig. S12). Also, labelling on the FSC plot has been improved. The new analysis also produced an average resolution of 1.96 Å. (figure below)

Overall resolution: We understand that the reported resolution being highest among published structures of ribosome, reviewer raised concern about possible overestimation of resolution. We also see the reviewer's comment about CryoSPARC and Relion. However, being the end user, we stick to much more user-friendly CryoSPARC, as it is a widely accepted cryo-EM data processing package with original publication in *Nature Methods* in 2017. Also, cryoSPARC – Relion comparison is beyond the capacity and interest of the current paper. Thus, we added Supplementary Fig. S15 to justify the high resolution obtained in our data.

Holes in the nucleotides and THB: In our THB-bound 70S structure we do observe clear holes in the densities of nucleotides as well as rRNA modifications (Supplementary Fig. S15A) (pasted below). We also see distinct EM-densities for amino acids side chain conformers (Fig. S15C), Mg⁺⁺ ion coordinated by water molecules (Fig. S15B) etc. justifying ~2Å resolution of our structure.

We do not see holes in the THB tetracycles due to relatively low resolution of the drug (Fig. S15D). We believe that comparatively low resolution of THB and the interacting rRNA bases may be due to greater conformational freedom of this region. Several rRNA bases around THB are dynamic e.g. A1492, A1493 and C1914, which do not have any definite binding partners. Also, since THB binding displaces the rRNA bases from their original positions, opposite force must also act on it from the rRNA bases to go back to the ground conformation. These are the probable reasons for relatively lower density of the drug and the rRNA bases around it.

Lower density of the drug is not uncommon. We draw the reviewer's attention to the 3.6 Å fusidic acid bound crystal structure of the EF-G-ribosome complex solved by Venki Ramakrishnan's group (Gao et al., *Science* 2009), where the authors write that – ‘We have placed the molecule (fusidic acid) in the most plausible orientation taking into account the shape of the density and the nature of potential interactions; however alternative orientations that are rotated or flipped by 180 degrees are also possible.’

Besides this, I have the following minor comments:

- In Fig. 3E the label at the top should read “and” instead of “aand”.

- **Response:** We thank reviewer for pointing out the typo. We now updated the labels in the figure.

- In Fig. 6D, the density for THB is hard to see. If solvent molecules coordinating the drug can be seen it is important to show them in this panel and, if possible, to avoid carving the density around the model.

- **Response:** We have updated the figure with no carved density around THB. Further the density of THB interacting to neighboring bases with solvent molecules is presented in Supplementary Fig. S15.

- On p. 18, it is stated that “The repositioned C1914 sterically interferes with the accommodating aa-tRNA in the A-site, explaining both impaired binding and increased drop-off in the presence of THB.” A structural comparison between the structure presented in this work and those of intermediates observed during A-tRNA delivery by EF-Tu and its subsequent accommodation (see studies from the Korostelev lab) would make a useful addition to Fig. 6 by giving the reader an idea of the extent of the clashes induced by the drug during aminoacyl-tRNA accommodation.

- **Response:** This concern of the reviewer has been properly addressed. See our response to comment 4 from reviewer 1.

- Similarly, on p. 20 - “The THB effect on termination originates most probably from THB-induced flipping of the monitoring bases A1492 and A1493 out of h44 (Fig. 6) which would impede RF binding and stop codon recognition since the flipped A1493 would clash with the domain-II of RF”. A panel showing potential clashes with RFs would be a good addition to Fig. 6.

- **Response:** We have also analyzed the clashes of RF1 with the flipped-out base C1914 from 23S rRNA presented in Fig. 6E. Also see our response to comment 4 from reviewer 1.

We have revised this statement according to our structural analysis. We now incorporate a figure depicting plausible clashes flipped-out base C1914 from 23S rRNA with the multiple residue of domain II of RF (Fig. 6E).

- Are there mutations that confer THB resistance? If so, where are they located and how does this fit with the proposed mechanism? A few lines to this effect could be added to the discussion section.

- **Response:** The resistance mutations for THB on bacterial ribosomes are not known yet. The possible reason could be that THB has not been into clinical use which might have prevented the possible evolution of resistance. Also, the THB binding site represents a critical and highly conserved region of the ribosome, mutations on the residues of this region would lead to lethal consequences for bacteria. Since our discussion is already long and elaborate due to biochemistry and structure part, we refrain to add speculative comments about THB resistance in discussion.

- In Supp. Fig. 1, could the authors comment on the different kinetics of subunit association in the absence of IFs with or without THB?

- **Response:** The variation in subunit association in Supplementary Fig. 1A, in which naked 50S and 30S (without IFs) were subjected to react is rather insignificant. The fast phases with or without THB (which form the main ground for the conclusion) are highly similar: The slight variation observed in the slow phase can be due to small difference in the subunit turnover in the presence of THB. The effect is however so small that it is hard to present as a stimulatory effect of THB on naked subunit association. That is why we write ‘THB also showed negligible effect on association of the naked 50S and 30S subunits (Supplementary Fig. S1A)’ (main text page 5).

- Heterogeneous 3D classification in Cryosparc yielded only two classes (Supp. Fig. 9): one consisting of 70S particles (96.1% of “good” particles) and the other predominantly of 50S particles (3.9%). Given the nature of the complex, one might also expect empty 70S ribosomes. Could the authors comment on the occupancy of fMet-tRNA^{fMet} in their complex and whether densities for the ribosome and tRNA appear to be stoichiometric?

- **Response:** We have taken the comment of the reviewer very carefully. To check whether we have empty 70S ribosome in our ‘70S particles’ classified by CryoSPARC, we ran an orthogonal SPA pipeline in Relion (Figure below). The first 3D classification using a global mask confirms that our reconstruction done using cryoSPARC was highly homogenous.

The class of 70S ribosomes containing fMet-tRNA^{fMet} (2,362,996 particles) comprised 90.43% of post 2D classification particles (2,613,018 particles). The proportion of 50S only particles (171,592 particles, 6.57%) and empty 70S ribosomes (78,430 particles, 3.0%) was significantly low. This analysis suggests that no defect exists with the heterogeneous 3D classification done in cryoSPARC. The high homogeneity can be attributed to our highly active and pure preparations of the ribosome and fMet-tRNA^{fMet} and stabilization by THB.

It is also important that they use the same contour levels to display densities for the tRNA, ribosome and mRNA in Fig. 6A and 6B, and that they use unmasked maps in which all density is shown for both of these panels (not just density around the model). This should be clearly specified in the figure legend.

- **Response:** The contour levels in Figure 6A and 6B were identical, the maps were unmasked. We have used the same contour levels for ribosome, tRNA, and mRNA in Fig. 6A, which has the model fitted to the excellent quality map. However, to convince the readers we also add a Supplementary Figure S13, which shows the density of these components at the same contour level in a magnified illustration. The figure legend has been written accordingly.

- A structural comparison with the earlier X-ray structure from the Steitz lab would be helpful, possibly as a supplementary figure. Did the gain in resolution reveal additional contacts or notable differences?

- **Response:** This is an important suggestion indeed. We have carefully compared THB binding and interactions in the earlier (Steitz gr) crystal structure with *Thermus thermophilus* ribosome and our cryo-EM structure with *E. coli* ribosome. We do see slight change in the orientation of the orthohydroxyphenyl group of THB (Supplementary Fig. S14 and page 13 main text, highlighted). The rest of the groups, rRNA bases and their interactions were similar.

The high resolution achieved in our cryo-EM however allowed us to unambiguously assign the orientation of the methyl groups of O8 and O9 of bound THB. In addition, it also showed rRNA modifications (both on 16S and 23S rRNAs) and protein side chains conformers (Supplementary Fig S15), all of which were not present in the earlier crystal structure. Furthermore, we could observe distinct EM-densities for water molecules coordinating Mg^{++} ions (Supplementary Fig. S15). We have thus modified the text and lifted up our observation in page 13 main text (highlighted). The results are summarized in the Supplementary Figs. S14 and S15.

REVIEWERS' COMMENTS

Reviewer #1 (Remarks to the Author):

The revised paper is much improved compared to the original submission and became even better. The authors have successfully addressed all my concerns.

Reviewer #2 (Remarks to the Author):

Dr. Sanyal and coworkers have done a comprehensive and thorough job addressing my comments and revising their manuscript, entitled "Antibiotic thermorubin tethers ribosomal subunits and impedes A-site interactions to perturb protein synthesis in bacteria". The expanded description of the kinetic experiments and the inclusion of the derivations and equations that were used to analyze and interpret the data have made the revised manuscript more accessible and able to stand on its own. The revised manuscript is consequently greatly improved and ready to be published in Nat Commun.

Reviewer #3 (Remarks to the Author):

The authors have fully and satisfactorily addressed all of my comments and any issues raised. I would therefore like to congratulate them for a very thorough and beautiful mechanistic study of thermorubin action.

2023-01-18

Point-by-point response to the referees

We thank all the three reviewers for accepting the revised version of the manuscript. We also thank them for doing a wonderful job for improvement of the manuscript.

Best regards,

Suparna

Uppsala University, Sweden